# Optimization Design and Test Bed of Fuzzy Control Rule Base for PV System MPPT in Micro Grid

**Jong-Chan Kim** [1] **, Jun-Ho Huh** [2,*] **and Jae-Sub Ko** [3,*]

1   Department of Computer Engineering, Sunchon National University, 255 Jungang-ro,
    Suncheon-city Jeollanam do 57922, Korea; seaghost@sunchon.ac.kr
2   Department of Data Informatics, Korea Maritime and Ocean University, Busan 600-716, Korea
3   Department of Electrical Engineering, Sunchon National University, 255 Jungang-ro,
    Suncheon-city Jeollanam do 57922, Korea
*   Correspondence: 72networks@kmou.ac.kr (J.-H.H.); kokos22@sunchon.ac.kr (J.-S.K.)

**Abstract:** This paper presents an optimal design of a fuzzy control rule base for tracking the maximum power point of a photovoltaic (PV) system. Fuzzy control is used for the maximum power point tracking (MPPT) of PV systems because it has the advantage of processing nonlinear systems. The rule base of fuzzy control depends on the user or designer's experience and determines the fuzzy control's performance. In this paper, we divide the MPPT state of the PV system into four cases according to the operating conditions, and propose the rule base design of the fuzzy control according to each case. The proposed method in the paper tests the MPPT performance using artificial lighting and compares the results with the conventional control method (proportional and integral (PI) and perturbation & observation (P&O) method) to prove its effectiveness.

**Keywords:** PV system; fuzzy control; MPPT; rule base; error; changing error; micro grid

## 1. Introduction

Energy is a very important factor in our lives. Conventional fossil fuels are finite and are characterized by problems with pollutants [1]. For these reasons, the demand for new energy sources is increasing. Solar energy is eco-friendly energy that is infinite, reusable, and free of pollutants [2]. The photovoltaic (PV) system, which uses solar energy to generate electricity, is most widely used to replace conventional fossil fuels [3,4]. The existing supply of electric energy has been a one-sided way of supplying electricity through large power plants, but the interest in smart grids, where information is exchanged between power plants and consumers, has also been increasing recently. Activating these smart grids requires spreading small-scale distributed power and building micro grids. Nowadays, in the energy market, small-scale distributed power sources based on renewable energy are spreading, and installation of the micro grid is continuing as a result [5–7].

The PV system is a system that converts solar energy into electricity, and its efficiency depends on the environmental conditions of solar radiation and the solar cell's temperature. The solar cells that make up the PV system have the semiconductor structure of P–N connection, and they are similar to the reverse saturation characteristics of the diode. Due to these characteristics, the solar cell has the characteristic of a non-linear change in current according to the operating voltage [8], and PV systems have maximum power at any particular operating point, which is called the maximum power point (MPP). Improving the efficiency of the PV system requires constant control so that it can be driven at the MPP.

The constant voltage (CV) method uses the maximum power voltage ($V_{mpp}$) measured at standard test condition (STC) or the specified voltage calculated [9–11], and the open circuit voltage (OCV)

method is controlled at voltage of 70–80% of the open circuit voltage [12–14]. The perturbation and observation (P & O) method uses the relationship between voltage and power in the PV module [15–17]. The voltage is perturbed, and the corresponding change in power is observed to track the maximum power point by varying the voltage in the direction of increasing the power. The incremental conductance (IncCond) method uses the slope of the power-voltage (P–V) curve, which represents the relationship between the power ($P_{pv}$) and the voltage ($V_{pv}$) of the solar cell [18–20]. The voltage is increased when the slope is positive and decreased when the slope is negative, becoming the maximum power point at zero.

The CV method has the disadvantage of not being the correct MPP for the current solar radiation, and the OCV method has the disadvantage of having to measure the open voltage whenever the solar radiation changes. Therefore, the P & O and IncCond methods are mostly used primarily for the MPPT control of the PV system [21]. These methods have a problem of oscillating from side to side at the maximum power point because they track the maximum power point by increasing or decreasing voltage or current in the given order. It also has a limit in improving tracking speed and in tracking accuracy because it varies the voltage or current to a fixed size [22]. To solve this problem, methods using artificial intelligence control such as PI control [1,23–25], fuzzy control [26–28], and neural network [29,30] were presented. PI control has a problem with fixed gain values, and neural networks suffer from poor response performance if sufficient learning is not achieved.

Neural networks use multiple layers as the number of variables increases, which increases the amount of data. In order to process the increased data, a high-performance processor with parallel processing capability is required, and there are no rules for configuring the input-hidden-output layer of the neural network, so it relies heavily on the user's experience of designing. In addition, if sufficient learning is not achieved, there is a problem that the response performance is deteriorated. In recent years, the field of utilizing neural networks is increasing and the amount of data is rapidly increasing accordingly, but the processing power of the processor cannot keep up with it. When a neural network is used in a control system, it causes an increase in the price of the system and has a disadvantage that does not guarantee sufficient stability [31,32].

The MPPT method of PV system using fuzzy control has been studied in various ways. Reference 28 is a representative of these methods. Reference 28 controls the MPPT of photovoltaic power generation using fuzzy control and compares the performance by the P & O method and simulation of MATLAB/Simulink. The reference 28 uses a rule base for fuzzy control, but does not propose how to design the used rule base. The control performance of the system using fuzzy control is greatly influenced by the rule base. In addition, the design of the rule base of fuzzy control depends on the user's experience [33]. Therefore, even in the same system, the design of the rule base varies depending on the user or designer, and different control results appear accordingly. In order to solve this problem, this paper classifies the PV system operation into four types and proposes a rule base design method that can obtain optimal control performance according to each operation state. The MPPT of the PV system is controlled with the rule base designed according to the method suggested in the paper, and the results are compared with the conventional P & O and PI methods and the results are analyzed.

The rest of this paper is organized as follows: Section 2 describes the proposed MPPT method, Section 3 compares and analyzes the MPPT control characteristics of the proposed method in this paper and the conventional method through experiment; finally, Section 4 presents the conclusion.

## 2. Proposed MPPT Method

In this paper, the maximum power point of the PV system is tracked using fuzzy control, which has strength in a nonlinear system and advantage of not requiring accurate modeling of the system. In using fuzzy control, the rule base and membership functions have a very significant impact on performance. Therefore, it is very important to design this. The most important part of the fuzzy theory developed by Zadeh in 1965 is the fuzzy rule base system [34]. Fuzzy systems have been successfully applied to problems that represent uncertainty and ambiguity because they use ambiguous rules

instead of general logic rules [35–38]. The design of the rule base is the most important variable in determining control performance. Therefore, there should be clear rules for designing rule bases. The conventional MPPT method used only the rule base, but no design process was presented [26–28].

In this paper, fuzzy control is used to track the maximum power point of the PV system, and the rule base used for this fuzzy control is designed by the procedure below. Equations (1) and (2) represent two input values of fuzzy control.

$$E(k) = \frac{P_{pv}(k) - P_{pv}(k-1)}{V_{pv}(k) - V_{pv}(k-1)} \tag{1}$$

$$CE(k) = E(k) - E(k-1) \tag{2}$$

Figure 1 shows the area of fuzzy rule base determined by error (E) and changing error (CE) as input of fuzzy control. Rule bases can be divided into four areas according to errors and changing errors as shown in Equations (3)–(6).

$$Zone\ 1\ :\ Error\ (E)\ is\ Negative\ AND\ Changing\ Error\ (CE)\ is\ Negative \tag{3}$$

$$Zone\ 2\ :\ Error\ (E)\ is\ Positive\ AND\ Changing\ Error\ (CE)\ is\ Negative \tag{4}$$

$$Zone\ 3\ :\ Error\ (E)\ is\ Negative\ AND\ Changing\ Error\ (CE)\ is\ Positive. \tag{5}$$

$$Zone\ 4\ :\ Error\ (E)\ is\ Positive\ AND\ Changing\ Error\ (CE)\ is\ Positive. \tag{6}$$

|  |  | Error(E) | |
| --- | --- | --- | --- |
|  |  | Negative | Positive |
| Changing Error(CE) | Negative | Zone 1 | Zone 2 |
|  | Positive | Zone 3 | Zone 4 |

**Figure 1.** Rule base area according to error (E) and changing error (CE).

Figure 2 shows the P–V characteristic curve of the PV system. Error (E) is equal to the slope of the P–V curve in Figure 2 and can be divided into the left and right sides of the maximum power point (MPP) according to the magnitude of the error. In order to track the maximum power point on the left side of the maximum power point, the voltage must be increased; on the right side, the maximum power point can be tracked by reducing the voltage.

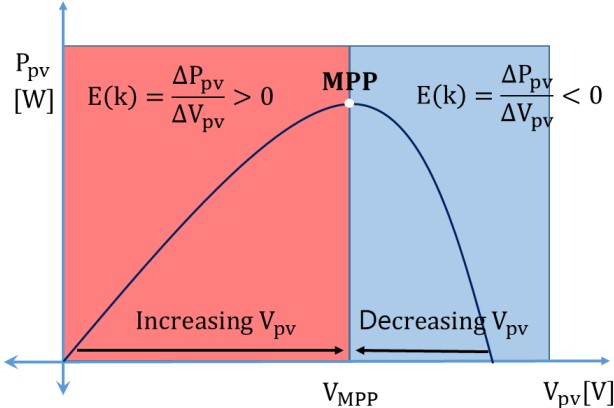

**Figure 2.** Maximum Power Point Tracking direction according to error (E).

Figure 3 shows the characteristics of tracking the maximum power point to the left of the maximum power point. Case 1 is tracking in the direction of the maximum power point from the left side, Case 2 is a case of passing the maximum power point. Since the error is equal to the slope of the P–V characteristic curve, the size decreases closer to the maximum power point and approaches zero. The characteristics shown in Case 1 and Case 2 are shown in Equations (7)–(12).

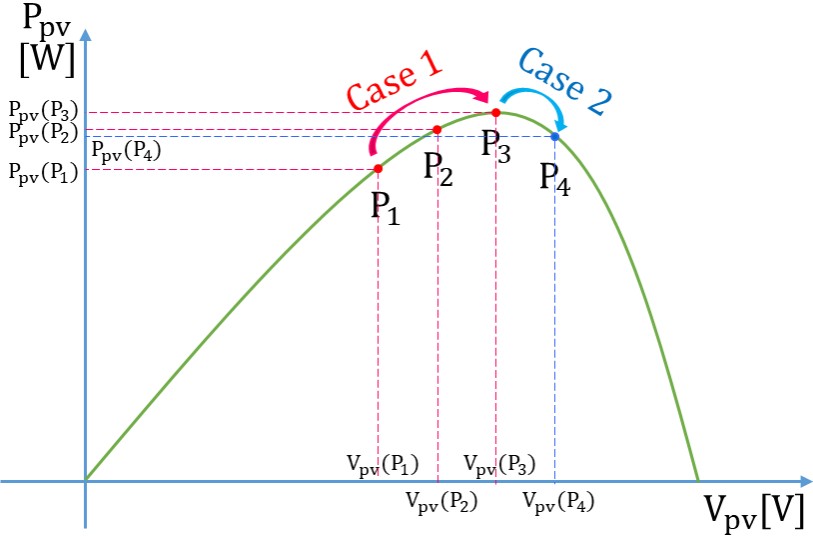

**Figure 3.** Maximum power point tracking characteristics (left side of maximum power point).

Case 1 : $P_1 \rightarrow P_2 \rightarrow P_3$, $P_{pv}(P_1) < P_{pv}(P_2) < P_{pv}(P_3)$, $V_{pv}(P_1) < V_{pv}(P_2) < V_{pv}(P_2)$.

$$E(P_3) = \frac{P_{pv}(P_3) - P_{pv}(P_2)}{V_{pv}(P_3) - V_{pv}(P_2)} = \frac{\text{Positive}}{\text{Positive}} = \text{Positive} \tag{7}$$

$$\left| E(P_3) \right| < \left| E(P_2) \right|, \ CE(P_3) = E(P_3) - E(P_2) = \text{Negative} \tag{8}$$

$$\text{Case 1 : E is Postive AND CE is Negative} \rightarrow \text{Zone 2} \tag{9}$$

Case 2 : $P_3 \rightarrow P_4$, $P_{pv}(P_3) > P_{pv}(P_4)$, $V_{pv}(P_3) < V_{pv}(P_4)$

$$E(P_4) = \frac{P_{pv}(P_4) - P_{pv}(P_3)}{V_{pv}(P_4) - V_{pv}(P_3)} = \frac{\text{Negative}}{\text{Positive}} = \text{Negative} \tag{10}$$

$$CE(P_4) = E(P_4) - E(P_3) = \text{Negative} - \text{Positive} = \text{Negative} \tag{11}$$

$$\text{Case 2 : E is Negative AND CE is Negative } \rightarrow \text{ Zone 1} \tag{12}$$

According to Equations (7) and (8), Case 1 corresponds to Zone 2 in Figure 1, and Case 2 represents Zone 1. Because Zone 2 is Case 1 that tracks toward the MPP from the left side of the MPP, the voltage should be increased to track the maximum power point, and Zone 1 corresponding to Case 2 should be reduced because it exceeds the MPP. To improve the performance of tracking the maximum power point, it must be tracked quickly and accurately. Large control values are required for fast tracking speeds, and small control values are needed for accurate tracking. When it is far from the maximum power point, the control value should be set large because the slope and E are large; the closer to the maximum power point, the smaller the slope and error, so the control value should be set smaller. Since the error is reduced as they are closer to the MPP, the changing error is also reduced.

In Zone 2, the rule base should be designed such that the control amount gradually decreases as error and changing error approach zero (ZE). Figure 4 shows the rule base design under these conditions. The magnitude of the control amount can be divided into three stages: B (Big), M (Medium), and S (Small). As the maximum power point is approached, the control amount gradually decreases from B→M→S. The MPPT control of PV systems is achieved by DC-DC converters or inverters. DC-DC converters for MPPT control generally use Buck Converter and Boost Converter [39]. In Buck converter, the PV module's voltage has the characteristic of decreasing from the open circuit voltage, and the Boost converter's voltage is increased from zero. In this paper, MPPT control is performed using Buck Converter, and voltage increases as the duty ratio of the PWM (Pulse Width Modulation) decreases. Zone 2 must increase the voltage for maximum power point tracking, so it must be controlled in the negative direction to reduce the duty ratio. Therefore, the control amount of the rule base of Zone 2 has the variables of negative big (NB), negative medium (NM), and negative small (NS).

| | | Error(E) | | | | | | |
|---|---|---|---|---|---|---|---|---|
| | | NB | NM | NS | ZE | PS | PM | PB |
| Changing Error(CE) | NB | PB | PB | PM | | NM | NB | NB |
| | NM | PB | PM | PS | | NS | NM | NB |
| | NS | PM | PS | PS | | NS | NS | NM |
| | ZE | | | | | | | |
| | PS | | | | | | | |
| | PM | | | | | | | |
| | PB | | | | | | | |

⟶ Maximum Power Point Direction

**Figure 4.** Rule base design for Zone 1 and Zone 2.

Zone 1 represents the case wherein the maximum power point is passed when tracking to the left of the maximum power point. Therefore, tracking in the direction opposite Zone 2 is necessary to track in the direction of maximum power point. In Zone 1, if error and changing error are close to ZE, it is designed to reduce the amount of control because it is close to the maximum power point. The Zone 1 rule base control amount must control the duty ratio in a positive direction to reduce voltage, and the control amount has positive big (PB), positive medium (PM), and positive small (PS).

Figure 5 shows the characteristics when tracing the maximum power point to the right of the maximum power point. To track in the direction of the maximum power point to the right of the

maximum power point, the voltage must be reduced. Case 3 shows the direction of tracking from the right side of the maximum power point to the maximum power point, and Case 4 is the case of passing the maximum power point. The characteristics of error and changing error in Cases 3 and 4 are presented in Equations (13)–(18).

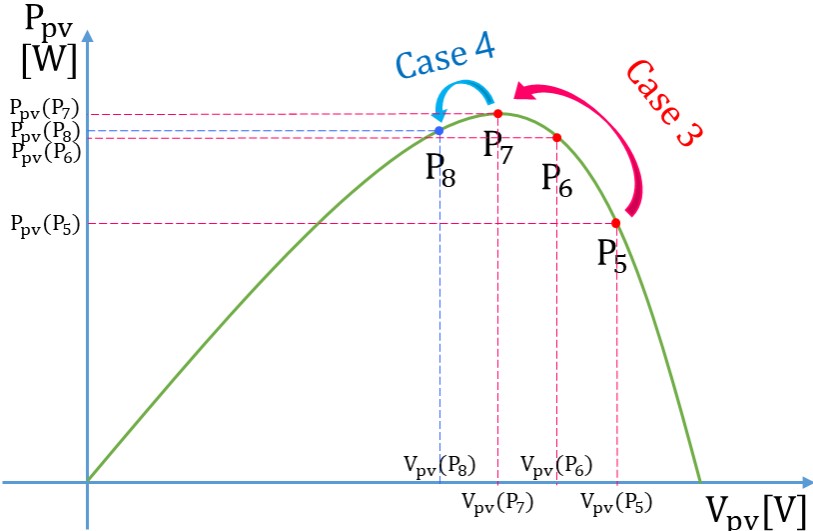

**Figure 5.** Maximum power point tracking characteristics (right side of maximum power point).

Case 3 : $P_5 \to P_6 \to P_7$, $P_{pv}(P_5) < P_{pv}(P_6) < P_{pv}(P_7)$, $V_{pv}(P_5) > V_{pv}(P_6) > V_{pv}(P_7)$.

$$(P_7) = \frac{P_{pv}(P_7) - P_{pv}(P_6)}{V_{pv}(P_7) - V_{pv}(P_6)} = \frac{\text{Positive}}{\text{Negative}} = \text{Negative} \tag{13}$$

$$\left|E(P_7)\right| < \left|E(P_6)\right| CE(P_7) = E(P_7) - E(P_6) = \text{Negative}(E(P_7)) - \text{Negative}(E(P_6)) = \text{Positive} \tag{14}$$

$$\text{Case 3 : E is Negative AND CE is Positive } \to \text{Zone 3} \tag{15}$$

Case 4 : $P_7 \to P_8$, $P_{pv}(P_7) > P_{pv}(P_8)$, $V_{pv}(P_7) > V_{pv}(P_8)$

$$E(P_8) = \frac{P_{pv}(P_8) - P_{pv}(P_7)}{V_{pv}(P_8) - V_{pv}(P_7)} = \frac{\text{Negative}}{\text{Negative}} = \text{Positive} \tag{16}$$

$$CE(P_8) = E(P_8) - E(P_7) = \text{Positive} - \text{Negative} = \text{Positive} + \text{Positive} = \text{Positive} \tag{17}$$

$$\text{Case 4 : E is Positive AND CE is Positive } \to \text{Zone 4} \tag{18}$$

When tracking the maximum power point to the right of the maximum power point, Case 3, controlled in the direction of the maximum power point, corresponds to Zone 3 in the rule base of Figure 1. To track the maximum power point in Case 3, the voltage must be reduced; in the case of Buck Converter, the duty ratio of the PWM must be increased. In Case 3, the slope of the P–V curve also approaches zero as the maximum power point is approached. Since the error gradually becomes zero, the changing error also decreases gradually to zero; when the error and changing error become zero, the operating point becomes the maximum power point. Since Zone 3 represents Case 3 in the rule base of Figure 1, the rule base can be designed as shown in Figure 6 to satisfy the error and changing error conditions of Case 3.

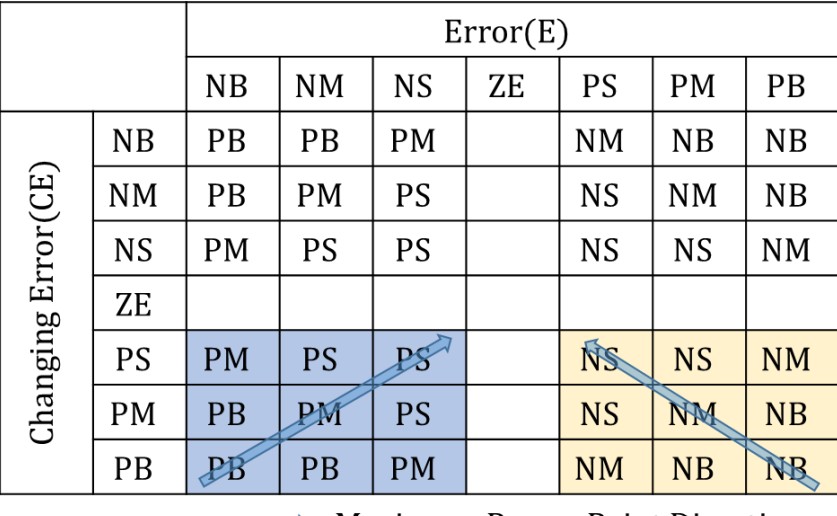

| | | NB | NM | NS | ZE | PS | PM | PB |
|---|---|---|---|---|---|---|---|---|
| | | \multicolumn Error(E) | | | | | | |
| | NB | PB | PB | PM | | NM | NB | NB |
| | NM | PB | PM | PS | | NS | NM | NB |
| | NS | PM | PS | PS | | NS | NS | NM |
| Changing Error(CE) | ZE | | | | | | | |
| | PS | PM | PS | PS | | NS | NS | NM |
| | PM | PB | PM | PS | | NS | NM | NB |
| | PB | PB | PB | PM | | NM | NB | NB |

Maximum Power Point Direction

**Figure 6.** Rule base design for Zone 3 and Zone 4.

As the error and changing error approach ZE in the rule base, it gradually changes from B→M→S to reduce the magnitude of the control amount. In Case 3, the voltage must be reduced, so the control value of the rule base is set to positive big (PB), positive medium (PM), and positive small (PS) to increase the duty ratio of PWM. As a case wherein the operating point passes the maximum power point when control starts to the right of the maximum power point and tracks the maximum power point, Case 4 corresponds to Zone 4 in Figure 1 due to error and changing error characteristics. Since Case 4 passed the maximum power point, when the voltage was reduced in Case 3, there is a need for control to increase the voltage to track the maximum power point.

In order to increase the voltage in Case 4, the duty ratio of PWM must be decreased, and the set value of the fuzzy rule base is used as the positive value. In Case 4, the error and changing error increase as you move away from the maximum power point, and the error and error change values decrease as you approach. Therefore, it is designed to reduce the control amount as the error and changing error are closer to the ZE of rule base. Figure 6 shows the rule base design of Zone 4 for Case 4.

In the fuzzy rule base of Figure 1, when the error is ZE, the slope of the P–V curve becomes zero, so the operating point becomes the maximum power point. Therefore, when the error is ZE, the rule base can be expressed as shown in Figure 7.

| | | NB | NM | NS | ZE | PS | PM | PB |
|---|---|---|---|---|---|---|---|---|
| | | \multicolumn Error(E) | | | | | | |
| | NB | PB | PB | PM | ZE | NM | NB | NB |
| | NM | PB | PM | PS | ZE | NS | NM | NB |
| | NS | PM | PS | PS | ZE | NS | NS | NM |
| Changing Error(CE) | ZE | | | | ZE | | | |
| | PS | PM | PS | PS | ZE | NS | NS | NM |
| | PM | PB | PM | PS | ZE | NS | NM | NB |
| | PB | PB | PB | PM | ZE | NM | NB | NB |

**Figure 7.** Rule base design when the magnitude of error is zero.

In order for the changing error to be ZE, the previous error must have the same magnitude as the current error. Figure 8 shows the P–V curve of the PV system; when it is far from the maximum power point, the P–V curve appears almost straight. Therefore, since the slope is constant, the changing error can be zero as shown in Equation (20).

$$P_{pv}(k+1) - P_{Pv}(k) \cong P_{pv}(k) - P_{pv}(k-1), \ V_{pv}(k+1) - V_{pv}(k) = V_{pv}(k) - V_{pv}(k-1). \tag{19}$$

$$CE(k+1) = E(k+1) - E(k) = \frac{P_{pv}(k+1) - P_{pv}(k)}{V_{pv}(k+1) - V_{pv}(k)} - \frac{P_{pv}(k) - P_{pv}(k-1)}{V_{pv}(k) - V_{pv}(k-1)} \cong 0 = Zero \tag{20}$$

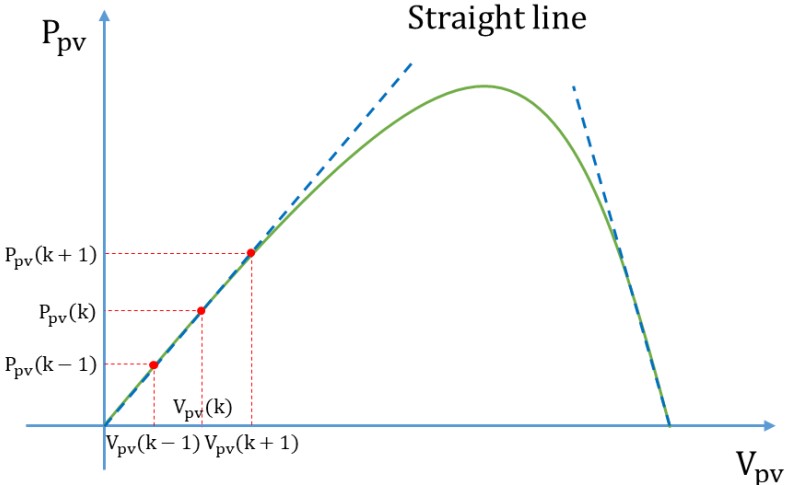

**Figure 8.** Straight line characteristics of the power-voltage (P–V) curve.

Figure 9 shows the condition under which solar radiation gradually increases when tracking the maximum power point. If the radiation is constant, the error decreases as the slope decreases near the maximum power point. If the radiation increases gradually, however, the power increases, and the previous and current errors can be equal. When radiation is Radiation B, the operating point changes from $P_9 \rightarrow P_{10} \rightarrow P_{11}$. If the radiation increases to Radiation A, however, the operating point is $P_{10}$ to $P_{12}$; at this time, the changing error is zero because of the characteristics shown in Equations (21) and (22).

$$P_{pv}(P_{10}) - P_{Pv}(P_9) \cong P_{pv}(P_{12}) - P_{pv}(P_{10}), \ V_{pv}(P_{10}) - V_{pv}(P_9) = V_{pv}(P_{12}) - V_{pv}(P_{10}) \tag{21}$$

$$CE(P_{12}) = E(P_{12}) - E(P_{10}) = \frac{P_{pv}(P_{12}) - P_{pv}(P_{10})}{V_{pv}(P_{12}) - V_{pv}(P_{10})} - \frac{P_{pv}(P_{10}) - P_{pv}(P_9)}{V_{pv}(P_{10}) - V_{pv}(P_9)} \cong 0 = Zero \tag{22}$$

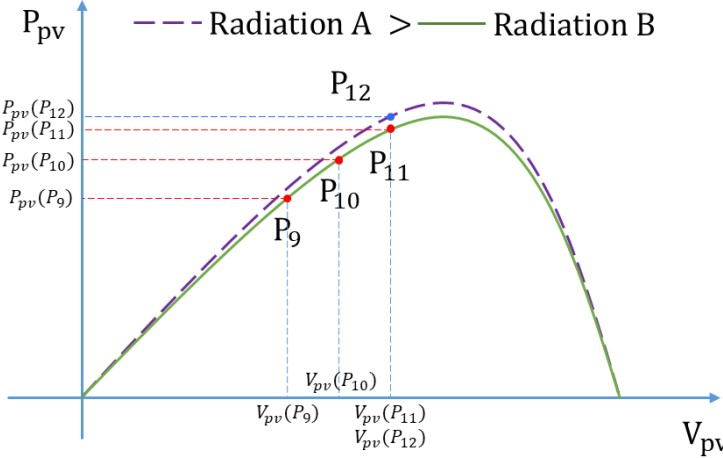

**Figure 9.** P–V curve characteristics for varying radiation conditions.

According to Figures 8 and 9, when the changing error becomes ZE, it is necessary to increase the control amount for tracking the maximum power point because of the great distance from the maximum power point or the increase in radiation. By designing a condition wherein the changing error becomes ZE under these conditions, the fuzzy control rule base shown in Table 1 is completed.

**Table 1.** Rule base for fuzzy control.

|  |  | Error (E) | | | | | | |
| --- | --- | --- | --- | --- | --- | --- | --- | --- |
|  |  | **NB** | **NM** | **NS** | **ZE** | **PS** | **PM** | **PB** |
|  | NB | PB | PB | PM | ZE | NM | NB | NB |
|  | NM | PB | PM | PS | ZE | NS | NM | NB |
|  | NS | PM | PS | PS | ZE | NS | NS | NM |
| Changing Error (CE) | ZE | PB | PM | PM | ZE | NM | NM | NB |
|  | PS | PM | PS | PS | ZE | NS | NS | NM |
|  | PM | PB | PM | PS | ZE | NS | NM | NB |
|  | PB | PB | PB | PM | ZE | NM | NB | NB |

Figure 10 shows the MPPT system presented in this paper. PV voltage ($V_{pv}$) and current ($I_{pv}$) are inputted through the voltage and current sensors, with the input values of the fuzzy controller calculated using Equations (1) and (2). The fuzzy controller is matched to a value between -1 and 1 through two gain values GE and GCE for fuzzification, and an output value ($\Delta D$) is calculated using a membership function and a rule base. Equation (24) shows the output value by the center of gravity method (COG). The calculated output value controls the duty ratio of PWM for MPPT control through defuzzification gain (GU).

$$e = \frac{E}{GE}, \ ce = \frac{CE}{GCE}. \tag{23}$$

$$\Delta d = \frac{\left(\sum_{j=1}^{n} \mu(d)_j \times (d)_j\right)}{\sum_{j=1}^{n} \mu(d)_j} \tag{24}$$

$$\Delta D = \Delta d \times GU \tag{25}$$

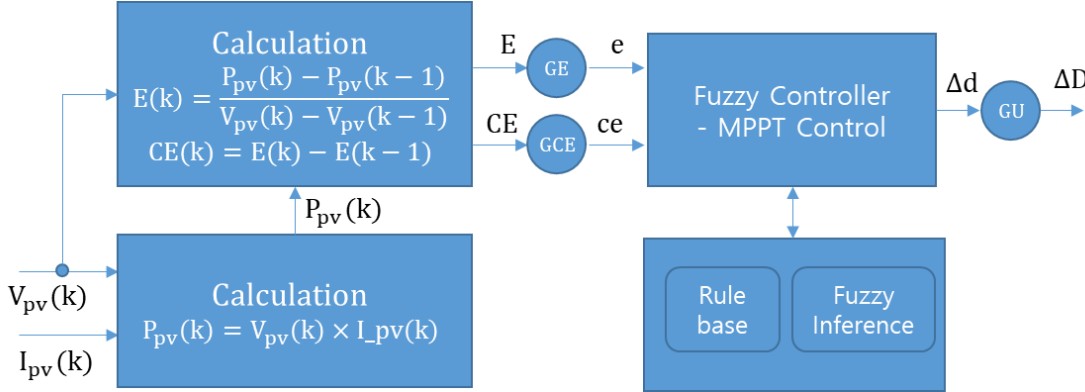

**Figure 10.** Maximum Power Point Tracking (MPPT) control using fuzzy control.

Figure 11 shows the membership function for error, changing error, and output.

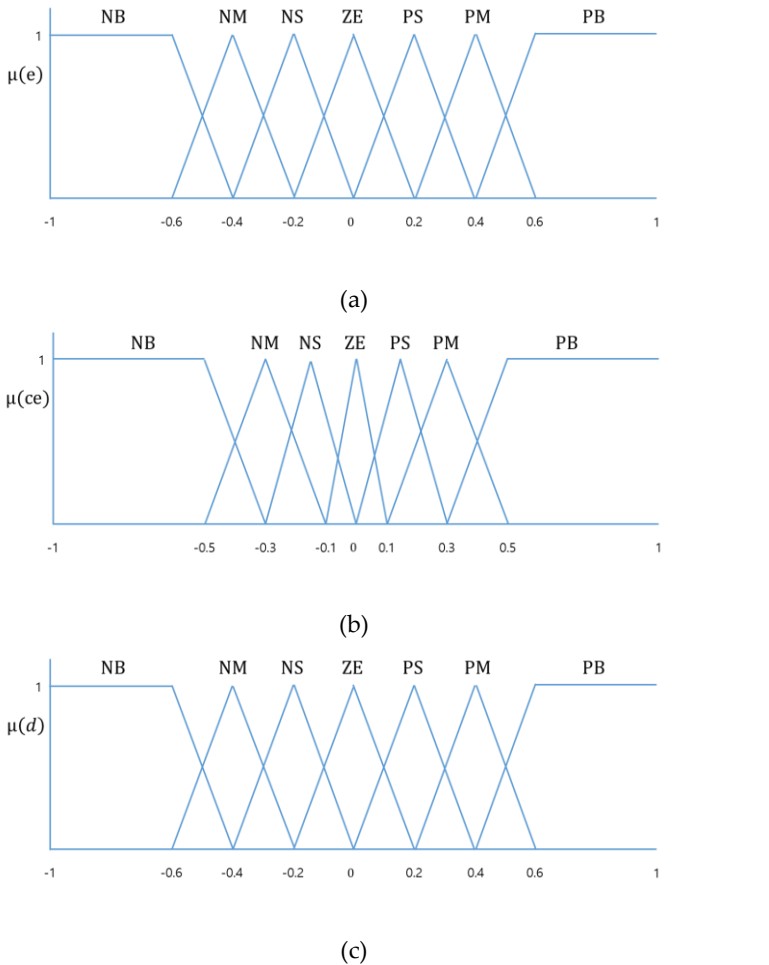

(a)

(b)

(c)

**Figure 11.** Membership function for fuzzy control: (**a**) error membership function; (**b**) changing error membership function; (**c**) output membership function.

## 3. Experiment Results

Figure 12 shows the circuit diagram and control system for testing the MPPT control performance of the PV system. In this paper, MPPT control is done by the buck converter, and voltage and current are measured using the INA219 voltage current sensor. Switching of the buck converter was performed

using the P-channel MOSFET (F9530N). The DC-DC Step Down Converter (KIS-3R33S) was used to maintain constant voltage and charge the cell phone battery.

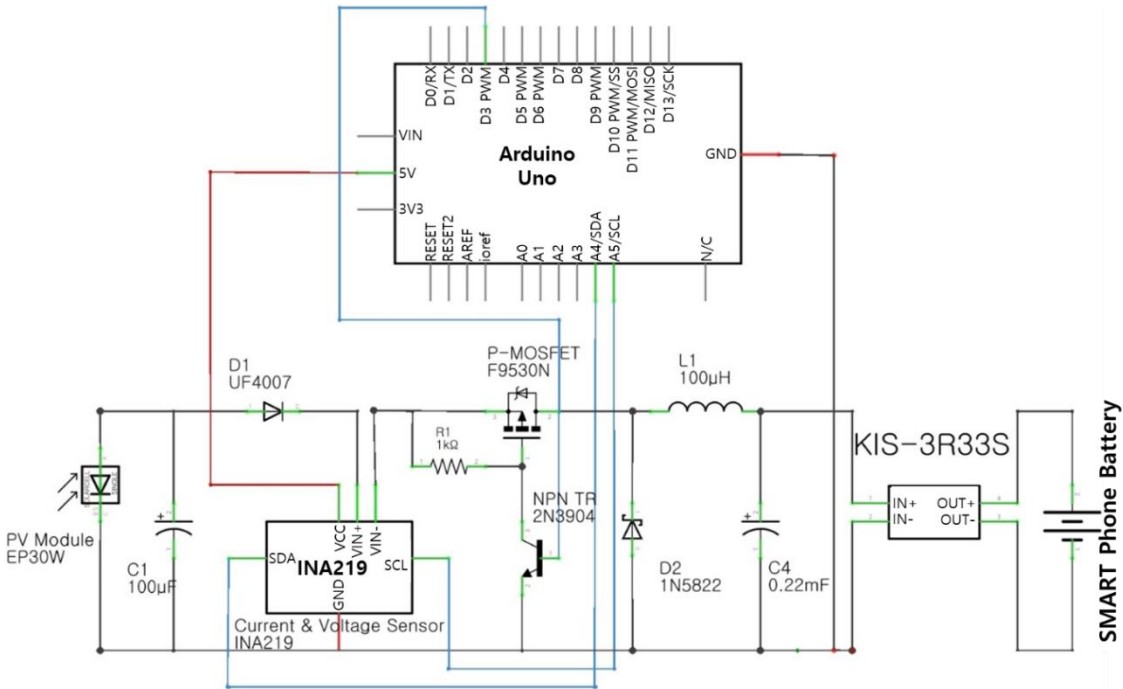

**Figure 12.** Experimental setup for the MPPT control performance test of the photovoltaic (PV) system.

Figure 13 shows the environment for the experiment. Figure 13a is a circuit for the experiment, and Figure 13b shows artificial light for constant radiation and PV module. In this paper, artificial lighting is used to test the MPPT performance of PV systems. Artificial lighting has lower radiation than sunlight, and for that reason, it produces very low power compared to the rated power [40]. And the output of the PV system is proportional to the amount of solar radiation when the temperature is constant [40,41]. Artificial lighting can maintain a constant radiation and can be appropriately changed by the user. Therefore, many studies using artificial lighting have been proposed to test the MPPT performance of PV systems in various environments [42,43]. The artificial light used in the paper is an incandescent lamp, and the incandescent lamp is the most suitable artificial light source for solar applications, especially in the case of crystalline silicon solar cells, the performance is improved over the AM1.5 spectrum [44]. The specifications of the PV module and smartphone battery used in the experiment are shown in Table 2. Mobile phones have a very close relationship with everyday life, and recently, PV systems have also been used to charge mobile phones in places such as public places and bus stops. Therefore, in the paper, the battery of the mobile phone was used as the load of the PV system. The radiation from the artificial light source was calculated using the output ratio. The rated output of the PV module used for the experiment is 30 W, which is a value measured at radiation 1000 W/m$^2$. In the paper, the output measured through the experiment using artificial lighting is about 3 W, and when calculated proportionally, the radiation is about 100 W/m$^2$.

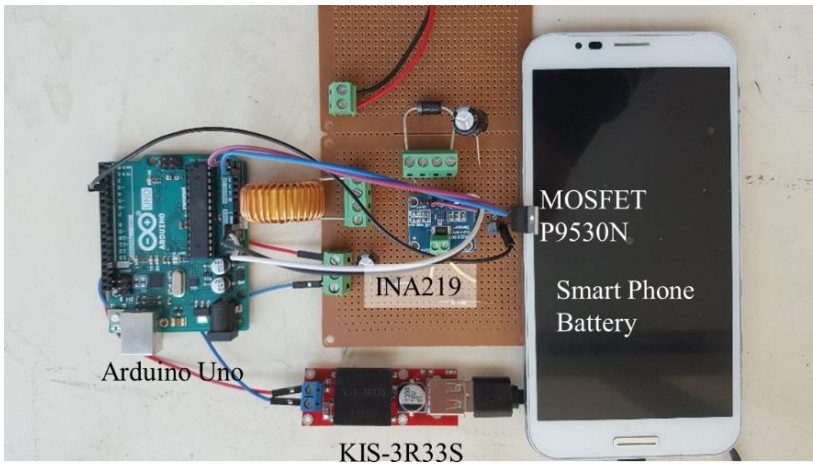

(a)

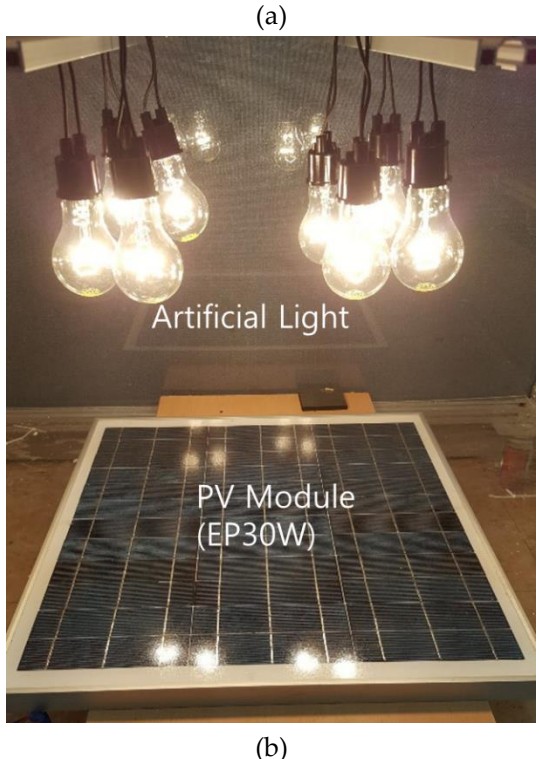

(b)

**Figure 13.** Environment for experiment: (**a**) Circuit for experiment; (**b**) Artificial light and PV module.

**Table 2.** Specifications of the photovoltaic (PV) module and Battery.

|  | Parameter | Value |
|---|---|---|
| PV Module | Maximum Power | 30 Wp |
|  | Open Circuit Voltage | 21 V |
|  | Optimum operation Voltage | 17.5 V |
|  | Short Circuit Current | 2 A |
|  | Optimum Operation Current | 1.7 A |
| Battery | Type | Li-ion |
|  | Capacity | 3200 mAh |
|  | Voltage | 3.7 V |

Figure 14 presents the characteristics of MPPT control using the fuzzy control proposed in this paper. Figure 14a shows the voltage and current of the PV module, Figure 14b, the power of the PV

module, Figure 14c, the output voltage, and Figure 14d, the duty ratio to control the DC-DC converter for maximum power point control and changing of duty ratio (ΔD) for PWM control. The maximum value of ΔD, which is a change in duty ratio, was set to 6.

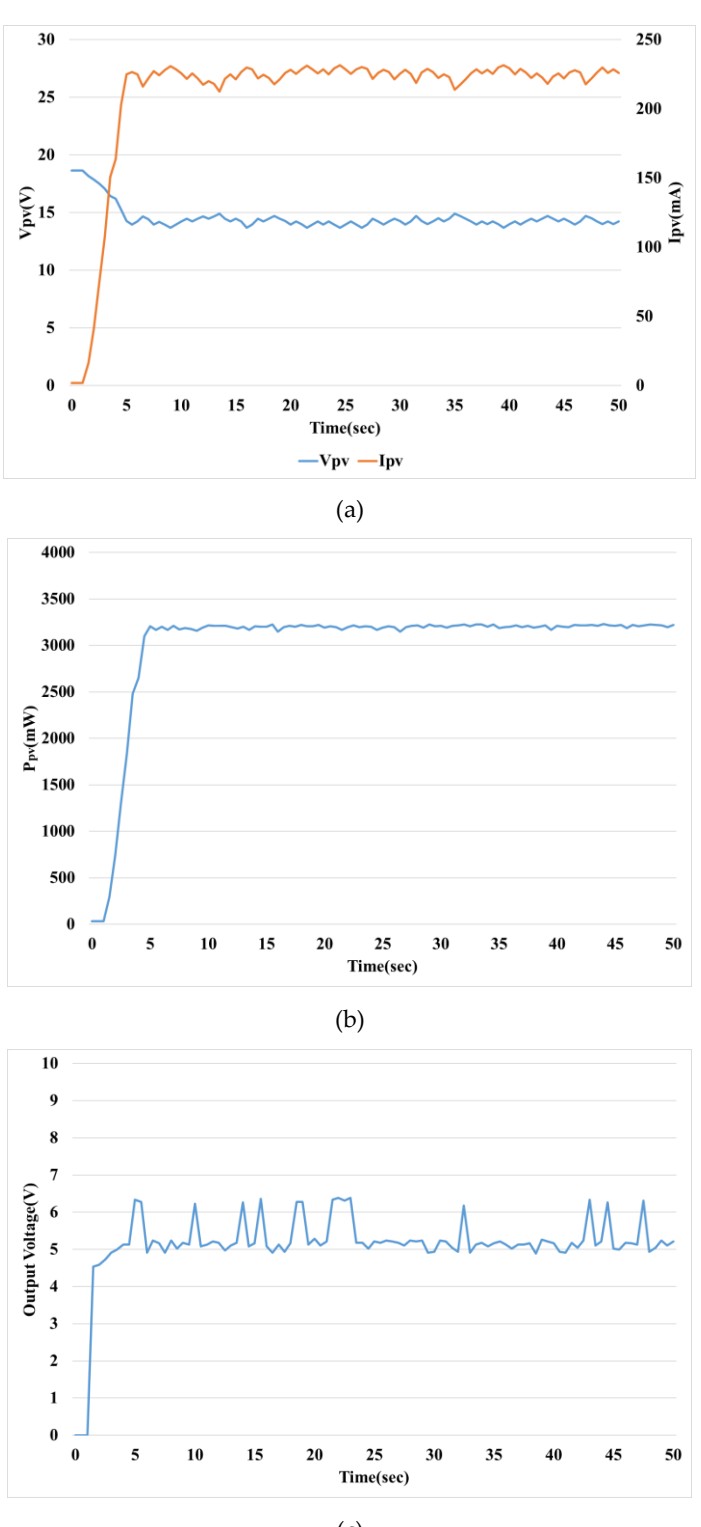

(a)

(b)

(c)

**Figure 14.** *Cont.*

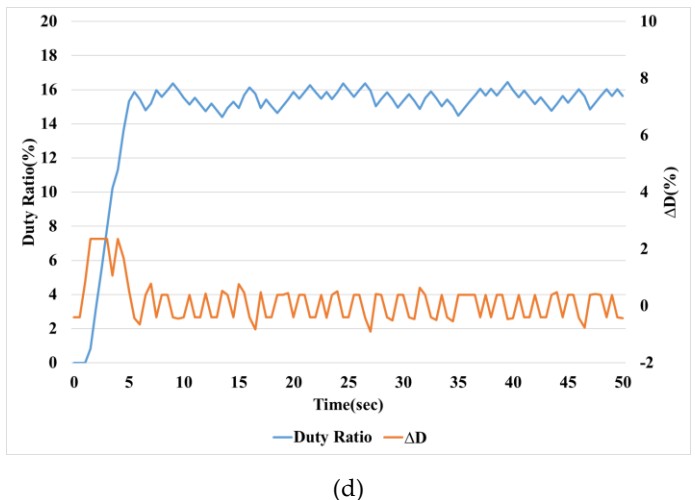

(d)

**Figure 14.** MPPT control characteristics (proposed Fuzzy MPPT method): (**a**) Voltage ($V_{pv}$) and Current ($I_{pv}$) of PV module; (**b**) Power ($P_{pv}$) of PV module; (**c**) Output Voltage; (**d**) Duty Ratio and $\Delta D$.

Figure 15 shows the control characteristics according to the error and changing error of the Fuzzy MPPT method presented in this paper. Figure 15a presents the error (E) and changing error (CE), and Figure 15b shows the enlarged Region A of Figure 15a. In Figure 15b, the operating points can be divided into 10 from (a) to (j). Figure 15c shows the control direction at 10 operating points. Cases 1 to 4 as proposed in this paper appear, with the maximum power point tracked according to the designed rule base. The characteristics appearing according to the error (E) and changing error (CE) at these 10 operating points are presented in Table 3. The operating points from (a) to (e) correspond to Case 3 in Figure 5. Case 3 tracks the maximum power point as the voltage decreases. The magnitudes of errors and changing errors become negative and positive, respectively, and become Zone 3 of the fuzzy rule base. In the designed rule base, Zone 3 has output $\Delta D$ of positive magnitude and decreases the voltage by increasing the PWM. The operating point (f) is a case wherein the maximum power point is passed while being controlled from the right side of the maximum power point, corresponding to Case 4 in Figure 5. In Case 4, both the error and the sign of the changing error are positive, becoming Zone 4 of the fuzzy rule base. Therefore, at operating point (f), the output becomes negative for the control of the maximum power point direction, and the voltage is increased by decreasing the PWM. The operating point (g) corresponds to Case 1 in Figure 7 because the control direction is the maximum power point direction by operating point (f). Since Case 1 tracks the maximum power point direction by increasing the voltage, output $\Delta D$ has a negative value because the PWM must be reduced. Operating points (h) and (i) are cases wherein the maximum power point is passed by the operation at the operating point (g), which becomes Case 2 in Figure 4. At this time, the error and the sign of the changing error are both negative, becoming Zone 1 of the fuzzy rule base. Zone 1 and Case 2 are directed toward the maximum power point when the voltage is reduced, so $\Delta D$ is positive. The operating point (j) exceeds the maximum power point by the operation of operating points (h) and (i), becoming Case 4 in Figure 5, and $\Delta D$ has a negative direction to increase the voltage.

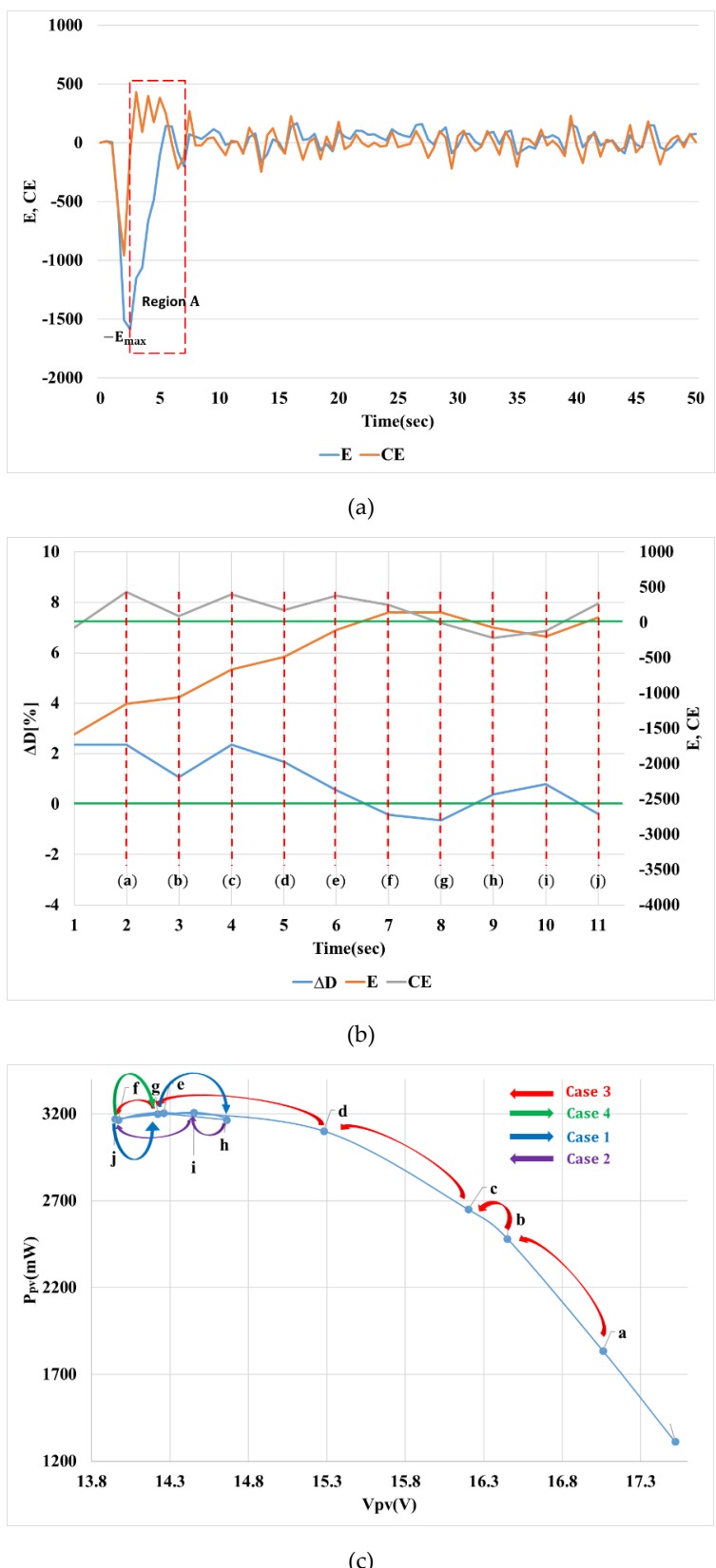

**Figure 15.** Control characteristics according to the error and changing error of the proposed Fuzzy MPPT method: (**a**) Error (E) and Changing Error (CE); (**b**) Region A Expansion; (**c**) Control direction.

**Table 3.** Control characteristics by operating point.

|  | (a) | (b) | (c) | (d) | (e) | (f) | (g) | (h) | (i) | (j) |
|---|---|---|---|---|---|---|---|---|---|---|
| E |  |  | Negative |  |  | Positive | Positive | Negative |  | Positive |
| CE |  |  | Positive |  |  | Positive | Negative | Negative |  | Positive |
| Zone |  |  | 3 |  |  | 4 | 2 | 1 |  | 4 |
| Case |  |  | 3 |  |  | 4 | 1 | 2 |  | 4 |
| ΔD |  |  | Positive |  |  | Negative | Negative | Positive |  | Negative |

Figures 16 and 17 show the MPPT control characteristics by the PI and P & O methods as the conventional control methods to compare the MPPT control characteristics. Figures 16 and 17a present the voltage and current of the PV module, Figures 16 and 17b,c,d show the power, output voltage and duty ratio and ΔD, respectively.

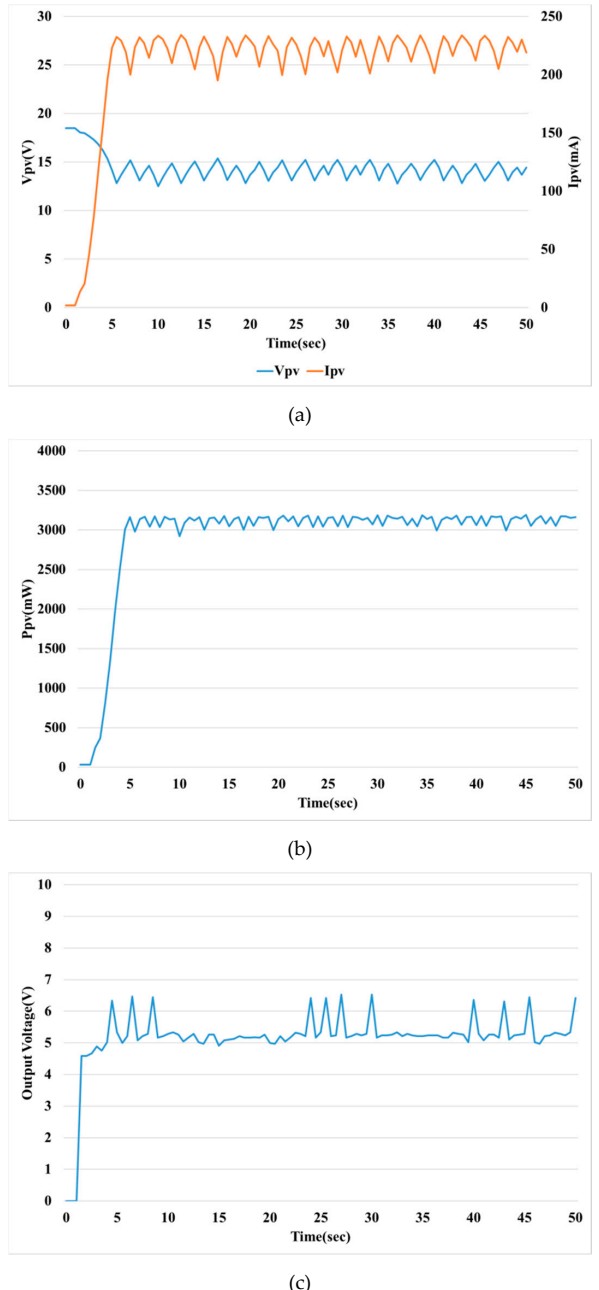

(a)

(b)

(c)

**Figure 16.** *Cont.*

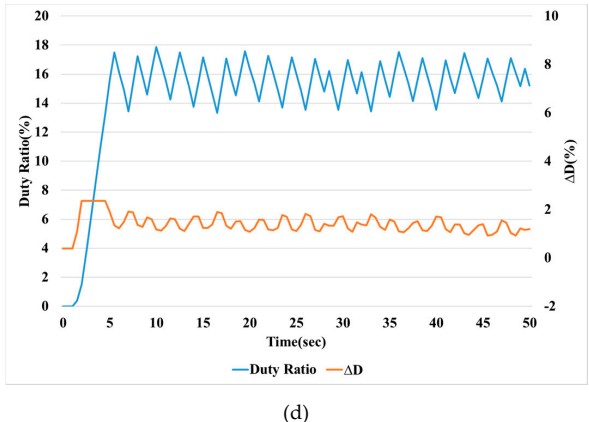

(d)

**Figure 16.** MPPT control characteristics (PI MPPT): (**a**) Voltage ($V_{pv}$) and Current ($I_{pv}$) of PV module; (**b**) Power ($P_{pv}$) of PV module; (**c**) Output Voltage; (**d**) Duty Ratio and $\Delta D$.

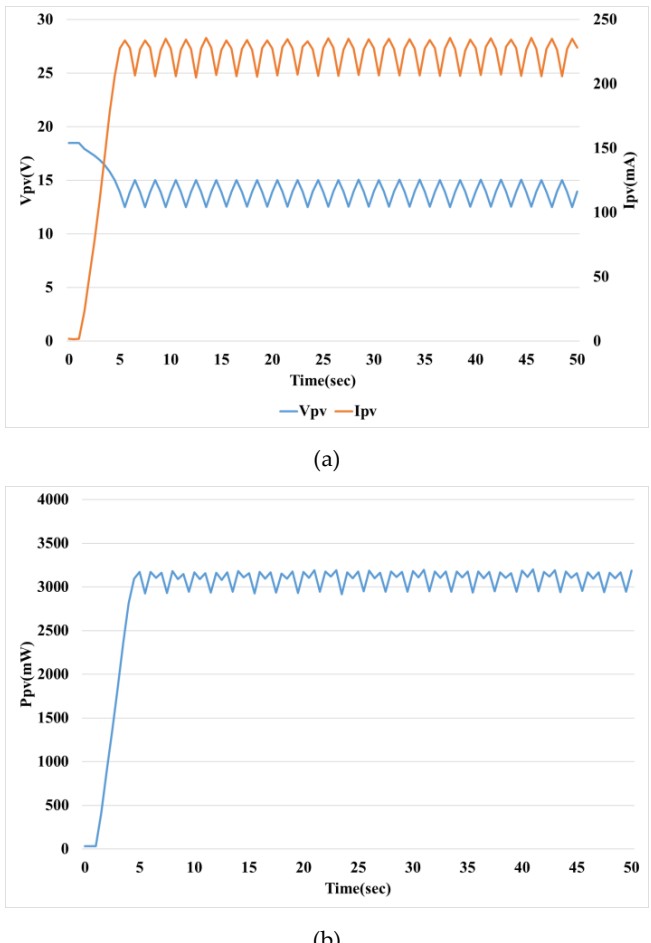

(a)

(b)

**Figure 17.** *Cont.*

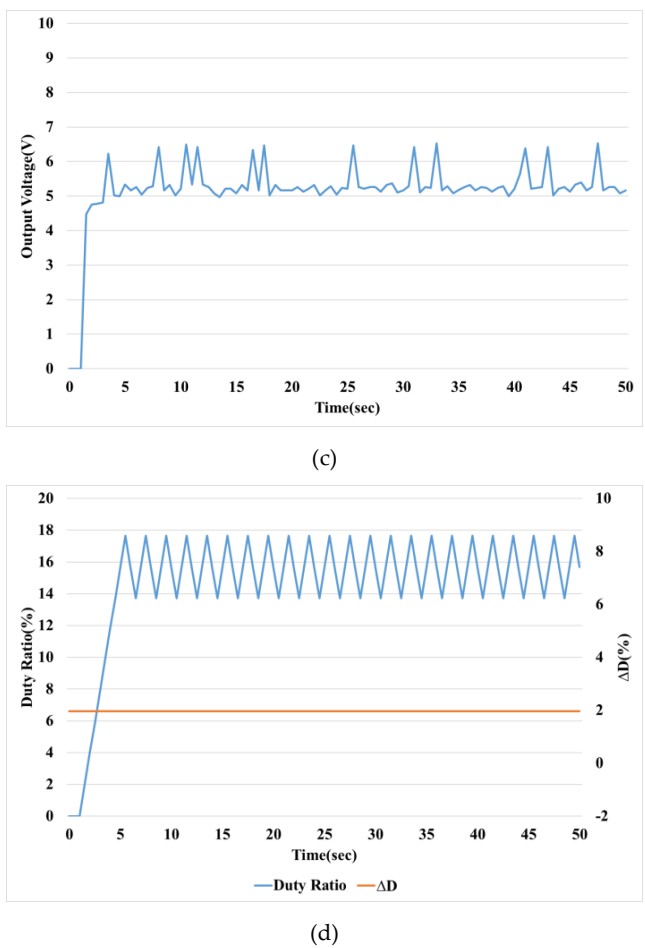

(c)

(d)

**Figure 17.** MPPT control characteristics (P & O MPPT): (**a**) Voltage ($V_{pv}$) and Current ($I_{pv}$) of PV module; (**b**) Power ($P_{pv}$) of PV module; (**c**) Output Voltage; (**d**) Duty Ratio and $\Delta D$.

Figure 18 shows the speed comparison for tracking the maximum power point in transient state. The gain value of the PI controller and the fixed $\Delta D$ value of the P & O method are set to be similar to the MPPT tracking speed of the Fuzzy control presented in this paper. The proportional gain ($K_p$) and integral gain ($K_i$) used in the PI controller are 0.005 and 0.00055, respectively, and the maximum value of $\Delta D$ controlling the PWM for maximum power tracking is set to 6. The fixed $\Delta D$ that controls the PWM of the P & O is set to 5.

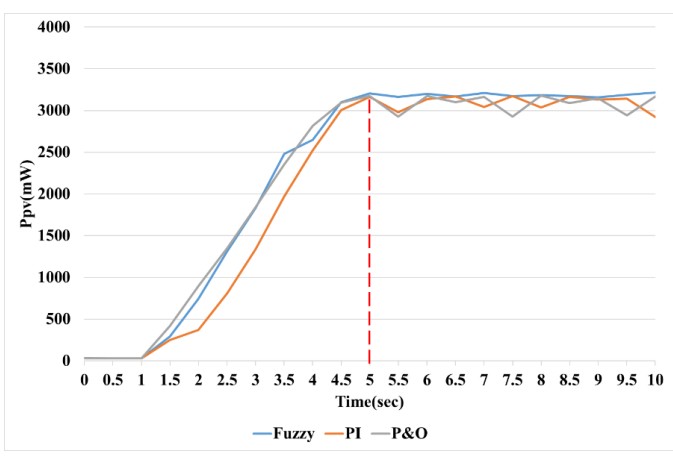

**Figure 18.** Comparison of MPPT response in transient state.

Figure 19 shows comparison of MPPT response characteristics in steady state. Figure 19a shows the Fuzzy MPPT method presented in this paper, and Figure 19b,c present the MPPT control by PI and P & O methods. The fuzzy MPPT method presented in this paper shows the smallest variation of voltage ($V_{pv}$) and power ($P_{pv}$) and achieves the most accurate MPPT control.

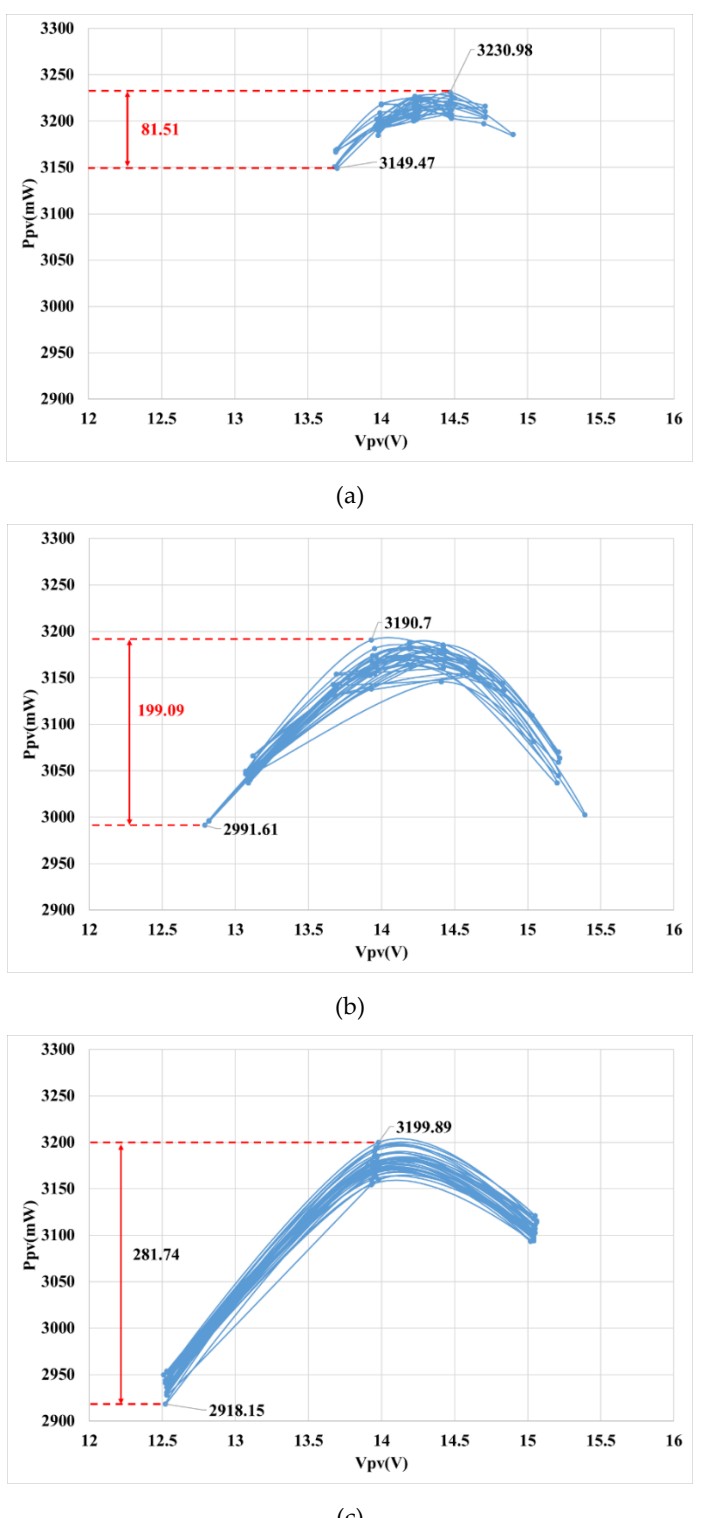

**Figure 19.** Comparison of MPPT response characteristics in steady state: (**a**) Proposed fuzzy MPPT method; (**b**) PI MPPT method; (**c**) P & O MPPT method.

Table 4 compares the characteristics in steady state of Figure 19 and the power ($P_{pv}$) and PWM values between 15 and 50 seconds. In terms of the peak to peak value of power, the PI and P & O MPPT methods showed 2.4 times and 3.4 times higher value, respectively, than the Fuzzy MPPT method presented in this paper. The Fuzzy MPPT method showed the least change in PWM.

**Table 4.** Comparison of steady state MPPT characteristics.

|  | Proposed Fuzzy MPPT | | PI MPPT | | P & O MPPT | |
|---|---|---|---|---|---|---|
|  | $P_{pv}$(mW) | PWM | $P_{pv}$(mW) | PWM | $P_{pv}$(mW) | PWM |
| MIN | 3149.47 | 36.93 | 2991.61 | 33.97 | 2918.15 | 35 |
| MAX | 3230.98 | 41.93 | 3190.7 | 44.82 | 3199.89 | 45 |
| Peak to Peak | 81.51 (100%) | 5 | 199.09 (244%) | 10.85 | 281.74 (345%) | 10 |
| Average | 3204.522 | 39.63 | 3125.989 | 39.77 | 3099.542 | 40.07 |

Figure 20 compares the response characteristics of power for changing the solar radiation conditions. Overall, the power ripple of the Fuzzy MPPT method is lowest. Table 5 shows a comparison of steady-state response for Figure 20. The power of three sections with increased radiation was compared, and the peak to peak value of power in all sections was lowest with the Fuzzy MPPT method.

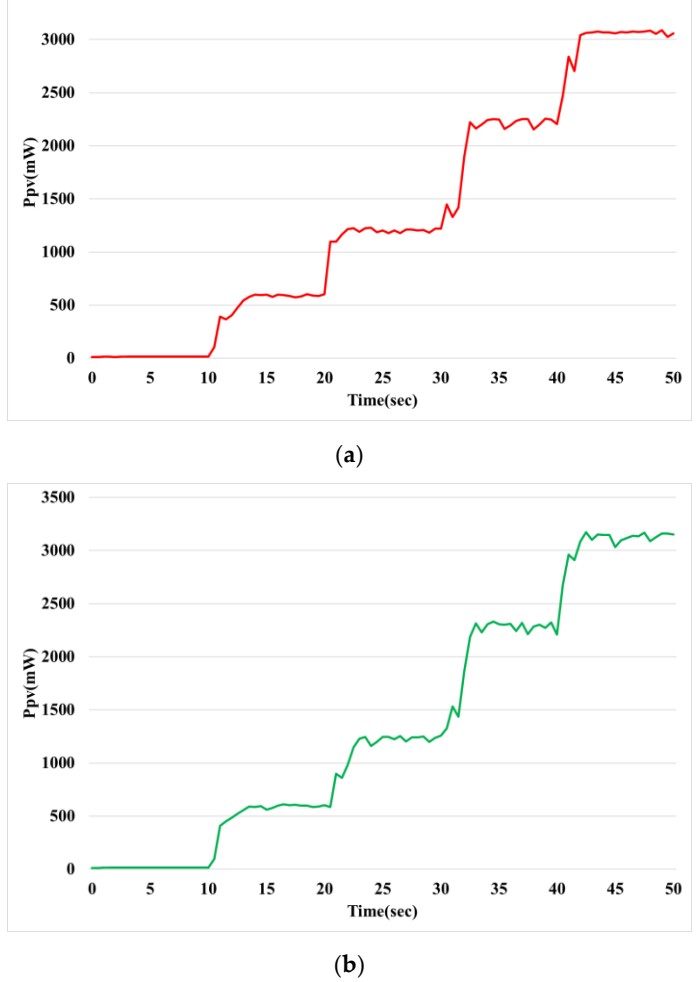

(a)

(b)

**Figure 20.** *Cont.*

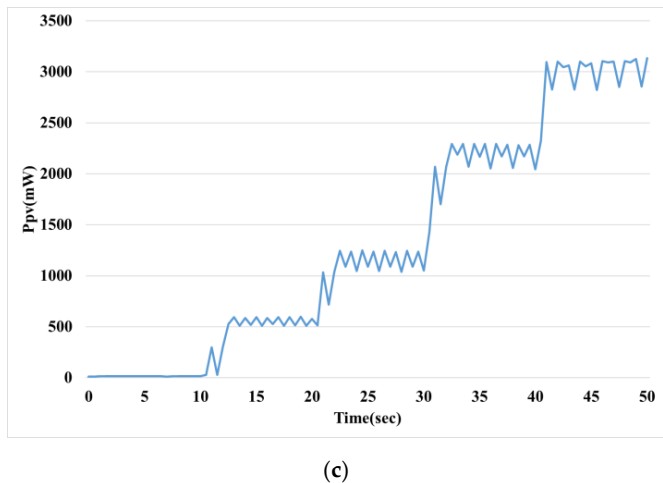

(**c**)

**Figure 20.** MPPT response characteristics for increasing solar radiation conditions: (**a**) Fuzzy MPPT method; (**b**) PI MPPT method; (**c**) P & O MPPT method.

**Table 5.** Comparison of steady state response characteristics by radiation.

|  |  | **Proposed Fuzzy MPPT** | **PI MPPT** | **P & O MPPT** |
|---|---|---|---|---|
| 23 (sec)–30 (sec) | MIN | 1177.58 | 1162.63 | 1040.12 |
|  | MAX | 1227.26 | 1257.53 | 1249.39 |
|  | Peak to Peak | 49.68 | 94.9 | 209.27 |
| 33 (sec)–40 (sec) | MIN | 2154.3 | 2209.3 | 2046.65 |
|  | MAX | 2254.42 | 2331.05 | 2295.53 |
|  | Peak to Peak | 100.12 | 121.75 | 248.88 |
| 43 (sec)–50 (sec) | MIN | 3027.11 | 3033.23 | 2824.27 |
|  | MAX | 3088.58 | 3167.68 | 3135.87 |
|  | Peak to Peak | 61.47 | 134.43 | 311.6 |

## 4. Conclusions

This paper presents the MPPT control of PV systems for micro-grid construction. PV systems are greatly affected by environmental conditions such as radiation and temperature. The output of the solar cell also varies nonlinearly according to the voltage-current relationship. Because the output changes according to the operating point, it is very important to operate the operating point at the point where the power is always maximized. Controlling the operating point to be the maximum power point is called MPPT.

The main methods used for MPPT are the P & O and IncCond methods. This method tracks the maximum power point according to a predetermined procedure. Since control is performed according to a predetermined order, however, there is a problem of control performance deteriorating when the radiation is suddenly changed and vibration is increased near the maximum power point. In addition, the general P & O and IncCond methods have a fixed amount of control, making it difficult to meet both tracking speed and tracking accuracy. To solve these problems, methods using PI, fuzzy control, and neural networks have been proposed. Among them, the PI control has a fixed gain value and a limitation of performance improvement. The neural network has a large amount of computation and a problem of performance being degraded if sufficient learning is not performed.

This paper has presented MPPT control using fuzzy control, which does not require accurate system modeling and has advantages in processing nonlinear systems. Fuzzy control inferences output values using rule base and membership function. Therefore, rule base and membership function have a great influence on fuzzy control performance. The existing fuzzy-based MPPT control does not present a rule base design but uses a rule base. Fuzzy control changes the response performance of the same system when the rule base is changed. Therefore, in this paper, we have proposed a method

of designing the rule base of the fuzzy control used in the PV system. When tracking the MPPT of the PV system, it can be divided into four cases according to the change of voltage and power: when the voltage rises, the power increases or decreases; when the voltage decreases, the power increases or decreases. The rule base used for fuzzy can be divided into four zones according to the error and changing error. The optimal rule base is designed by matching four cases that occur when tracking the maximum power point and four zones of the rule base.

The proposed controller compares control performance with conventional PI controllers and P & O methods for constant and varying radiation. The performance of MPPT control compares the tracking speed in transient state and the accuracy in steady state. The gain and setting values of all controllers were adjusted so that the tracking speed was similar, and the accuracy was compared in steady state. When the radiation is constant, the error values in the steady state are 81.51 mW, 199.09 mW and 281.74 mW for the proposed method, conventional PI method, and P and O method, respectively. The proposed method has the lowest error and the highest average power. The steady-state error was shown in the order of the proposed method <PI method <P & O method, even with increasing radiation conditions, and the tracking accuracy of the proposed method was the highest.

This paper has presented the optimal design of rule base of the fuzzy control used for the MPPT control of the PV system. The designed rule base of fuzzy control is designed for the case of constant radiation. Therefore, when the amount of insulation changes rapidly, it is necessary to study how to design the rule base.

**Author Contributions:** Conceptualization, J.-C.K., J.-H.H. and J.-S.K.; Data curation, J.-C.K. and J.-S.K.; Formal analysis, J.-C.K., J.-H.H. and J.-S.K.; Funding acquisition, J.-H.H.; Investigation, J.-C.K. and J.-S.K.; Methodology, J.-C.K., J.-H.H. and J.-S.K.; Project administration, J.-H.H. and J.-S.K.; Resources, J.-C.K., J.-H.H. and J.-S.K.; Software, J.-C.K., J.-H.H. and J.-S.K.; Supervision, J.-H.H. and J.-S.K.; Validation, J.-S.K.; Visualization, J.-S.K.; Writing—original draft, J.-C.K., J.-H.H. and J.-S.K.; Writing—review and editing, J.-H.H. and J.-S.K. All authors have read and agreed to the published version of the manuscript.

**Funding:** This work was supported by the National Research Foundation of Korea (NRF) grant funded by the Korean government (MSIT) (No.2017R1C1B5077157). Also, this research was supported by the Energy Cloud R&D Program through the National Research Foundation of Korea (NRF) funded by the Ministry of Science and ICT (NRF-2019M3F2A1073385).

**Conflicts of Interest:** The authors declare no conflict of interest.

## Abbreviations

| | |
|---|---|
| MPPT | Maximum Power Point Tracking |
| PV | PhotoVoltaic |
| P & O | Perturbation and Observation |
| PI | Proportional Integral |
| MPP | Maximum Power Point |
| CV | Constant Voltage |
| STC | Standard Test Condition |
| OCV | Open Circuit Voltage |
| IncCond | Incremental Conductance |
| P–V | Power–Voltage |
| $V_{pv}$ | Voltage of PV module |
| $I_{pv}$ | Current of PV module |
| $P_{pv}$ | power of PV module |
| $\Delta P_{pv}$ | Changing of $P_{pv}$ |
| $\Delta V_{pv}$ | Changing of $V_{pv}$ |
| PWM | Pulse Width Modulation |
| E | Error |
| CE | Changing Error |

NB　　　　　Negative Big
NM　　　　Negative Medium
ZE　　　　　Zero
NS　　　　　Negative Small
PB　　　　　Positive Big
PM　　　　Positive Medium
PS　　　　　Positive Small
GE　　　　　Gain of Error in fuzzy control
GCE　　　　Gain of Changing Error in fuzzy control
GU　　　　　Gain of output
ΔD　　　　　Changing of duty ratio
COG　　　　Center of Gravity

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
