# Peer review of "Optimization Design and Test Bed of Fuzzy Control Rule Base for PV System MPPT in Micro Grid"

_sustainability, doi:10.3390/su12093763_

Round 1
Reviewer 1 Report
The reviewed paper includes detailed optimization model of rule design for fuzzy control used for maximum power point tracking control of PV system. The detailed literaturÄ™ background is an introduction to author's own design rules for this kind of method. The article is clear and containes reach description, however two issues should be corrected:
- Poor visibility of descriptions in Fig. 15 (p.13).
- In paragraph no 5, p. 12, there are the results of tests / measurements described. However, there is no detailed information on the conditions of and operation parameters of the examined laboratory equipment (PV, baterries, radiation, etc.). It should be more clearly stated what is the base of comparison between the examined techniques, therefore this description must be improved.
Author Response
Comments and Suggestions for Authors
The reviewed paper includes detailed optimization model of rule design for fuzzy control used for maximum power point tracking control of PV system. The detailed literature background is an introduction to author's own design rules for this kind of method. The article is clear and containes reach description, however two issues should be corrected:
1.Poor visibility of descriptions in Fig. 15 (p.13).
-----
Reply -----
-----
First of all, thank you for reviewing our paper in detail and giving us an appropriate comment. The Co Corresponding author [1], First author [2], and Corresponding author [3] are majoring in Smart/Micro Grid theories for computer engineering. Although one of the drawbacks of the computer engineering-related studies is that the descriptions and explanations can be quite lengthy, but their advantage is that the contents of the proposal can be understood clearly just by reading them.
[1] Jun-Ho Huh
https://scholar.google.co.kr/citations?user=cr5wjNYAAAAJ&hl=ko
[2] Jong Chan Kim
https://scholar.google.co.kr/citations?user=YGPImkUAAAAJ&hl=ko
[3] Jae-Sub Ko
https://scholar.google.co.kr/citations?user=Vph8kvsAAAAJ&hl=ko
It was our intention to validate the proposed mathematical theories by performing simulations. At the same time, to cover the drawbacks, the contribution parts have been included in every possible section while correcting the contents with the help of a native English speaker to improve readability within a limited time frame. It seems that the Special Issue ‘Sustainable Building Retrofit and Energy Optimization [4]’, to which we’ve submitted our study, was to provide some understanding to the Sustainable engineers studying computerbuilding/energy/optimization engineering from the energy optimization point of view. Thus, we’ve included relevant discussions to make the paper more meaningful. The revised or added parts are being highlighted in red for your possible re-review.
-We improved the picture quality in Figure 15 based on the review's comments.
CHANGE 1)
Figure 15. Experimental setup for the MPPT control performance test of the PV system
In paragraph no 5, p. 12, there are the results of tests / measurements described. However, there is no detailed information on the conditions of and operation parameters of the examined laboratory equipment (PV, baterries, radiation, etc.). It should be more clearly stated what is the base of comparison between the examined techniques, therefore this description must be improved.
-----
Reply-----
-----
We added the environment for the experiment and the specifications of the PV module and battery used in the experiment according to the opinion of the Reviewer.
ADD 1)
Figure 16 shows the environment for the experiment. Fig. 16 (a) is a circuit for the experiment, and Fig. 16 (b) shows artificial light for constant radiation and PV module. The specifications of the PV module and smartphone battery used in the experiment are shown in Table 2, and radiation by artificial lighting is about 100 .
ADD 2)
(a)
(b)
Figure 16. Environment for experiment
(a) Circuit for experiment; (b) Artificial light and PV module
ADD 3)
Table 2.Specifications of the PV module and Battery
|
|
Parameter |
Value |
|
PV Module |
Maximum Power |
30 [Wp] |
|
Open Circuit Voltage |
21 [V] |
|
|
Optimum operation Voltage |
17.5 [V] |
|
|
Short Circuit Current |
2 [A] |
|
|
Optimum Operation Current |
1.7 [A] |
|
|
Battery |
Type |
Li-ion |
|
Capacity |
3200 [mAh] |

Reviewer 2 Report
the paper presented a fuzzy logic based MPPT method for PV system. however, many papers have been published for MPPT and fuzzy logic methods, and authors need to highlight new contributions of this research.
for example, a similar method has been published in 2014:
Y. Soufi, M. Bechouat, S. Kahla and K. Bouallegue, "Maximum power point tracking using fuzzy logic control for photovoltaic system," 2014 International Conference on Renewable Energy Research and Application (ICRERA), Milwaukee, WI, 2014, pp. 902-906.
Author Response
Comments and Suggestions for Authors
the paper presented a fuzzy logic based MPPT method for PV system. however, many papers have been published for MPPT and fuzzy logic methods, and authors need to highlight new contributions of this research.
for example, a similar method has been published in 2014:
- Soufi, M. Bechouat, S. Kahla and K. Bouallegue, "Maximum power point tracking using fuzzy logic control for photovoltaic system," 2014 International Conference on Renewable Energy Research and Application (ICRERA), Milwaukee, WI, 2014, pp. 902-906.
-----
Reply-----
-----
First of all, thank you for reviewing our paper in detail and giving us an appropriate comment. The Co Corresponding author [1], First author [2], and Corresponding author [3] are majoring in Smart/Micro Grid theories for computer engineering. Although one of the drawbacks of the computer engineering-related studies is that the descriptions and explanations can be quite lengthy, but their advantage is that the contents of the proposal can be understood clearly just by reading them.
[1] Jun-Ho Huh
https://scholar.google.co.kr/citations?user=cr5wjNYAAAAJ&hl=ko
[2] Jong Chan Kim
https://scholar.google.co.kr/citations?user=YGPImkUAAAAJ&hl=ko
[3] Jae-Sub Ko
https://scholar.google.co.kr/citations?user=Vph8kvsAAAAJ&hl=ko
It was our intention to validate the proposed mathematical theories by performing simulations. At the same time, to cover the drawbacks, the contribution parts have been included in every possible section while correcting the contents with the help of a native English speaker to improve readability within a limited time frame. It seems that the Special Issue ‘Sustainable Building Retrofit and Energy Optimization[4]’, to which we’ve submitted our study, was to provide some understanding to the Sustainable engineers studying computerbuilding/energy/optimization engineering from the energy optimization point of view. Thus, we’ve included relevant discussions to make the paper more meaningful. The revised or added parts are being highlighted in red for your possible re-review.
-We added example paper to the reference according to the opinion of the Reviewer, and added the difference between this paper and other papers using example paper and Fuzzy Control to the introduction.
ADD 1)
The MPPT method of PV system using fuzzy control has been studied in various ways. Reference 52 is a representative of these methods. Reference 52 controls the MPPT of photovoltaic power generation using fuzzy control and compares the performance by P&O method and simulation of MATLAB / Simulink. The reference 52 uses a rule base for fuzzy control, but does not propose how to design the used rule base. The control performance of the system using fuzzy control is greatly influenced by the rule base. In addition, the design of the rule base of fuzzy control depends on the user's experience [57]. Therefore, even in the same system, the design of the rule base varies depending on the user or designer, and different control results appear accordingly. In order to solve this problem, this paper classifies the PV system operation into four types and proposes a rule base design method that can obtain optimal control performance according to each operation state. The MPPT of the PV system is controlled with the rule base designed according to the method suggested in the paper, and the results are compared with the conventional P & O and PI methods and the results are analyzed.
ADD 2)
- Soufi, Y.; Bechouat, M.; Bouallegue, K. Maximum power point tracking fuzzy logic control for photovoltaic system. 2014 International Conference on Renewable Energy Research and Application(ICRERA), 2014 , 902-906.
ADD 3)
Figure 16 shows the environment for the experiment. Fig. 16 (a) is a circuit for the experiment, and Fig. 16 (b) shows artificial light for constant radiation and PV module. The specifications of the PV module and smartphone battery used in the experiment are shown in Table 2, and radiation by artificial lighting is about 100 .
ADD 4)
(a)
(b)
Figure 16. Environment for experiment
(a) Circuit for experiment; (b) Artificial light and PV module
ADD 5)
Table 2.Specifications of the PV module and Battery
|
|
Parameter |
Value |
|
PV Module |
Maximum Power |
30 [Wp] |
|
Open Circuit Voltage |
21 [V] |
|
|
Optimum operation Voltage |
17.5 [V] |
|
|
Short Circuit Current |
2 [A] |
|
|
Optimum Operation Current |
1.7 [A] |
|
|
Battery |
Type |
Li-ion |
|
Capacity |
3200 [mAh] |

Reviewer 3 Report
The present paper shows a new step in Photovoltaic electrical power generation centred in a new control system. In particular, the advantages of fuzzy logic versus PI and neural networks are showed and applied in a real case study.
The methodology is adequate, despite the fact that in my opinion, a little bit more information about how the fuzzy logic procedure was developed as it used to be done in neural networks.
Results are of interest due to the logical improve of power energy obtained. It may be of interest to explain the reason why a neuro fuzzy is not employed now and if it could be employed in a near future.
Finally, there are a lot of problems with equations in the whole paper so it must be revised its format (how to min in the *.pdf version). Other minor revision must be done at the first time that some abbreviations are employed (like PI) must be defined.
The main idea is of interest and really applied in the sustainability topic.
Author Response
Comments and Suggestions for Authors
The present paper shows a new step in Photovoltaic electrical power generation centred in a new control system. In particular, the advantages of fuzzy logic versus PI and neural networks are showed and applied in a real case study.
The methodology is adequate, despite the fact that in my opinion, a little bit more information about how the fuzzy logic procedure was developed as it used to be done in neural networks.
Results are of interest due to the logical improve of power energy obtained. It may be of interest to explain the reason why a neuro fuzzy is not employed now and if it could be employed in a near future.
-----
Reply -----
-----
First of all, thank you for reviewing our paper in detail and giving us an appropriate comment. The Co Corresponding author [1], First author [2], and Corresponding author [3] are majoring in Smart/Micro Grid theories for computer engineering. Although one of the drawbacks of the computer engineering-related studies is that the descriptions and explanations can be quite lengthy, but their advantage is that the contents of the proposal can be understood clearly just by reading them.
[1] Jun-Ho Huh
https://scholar.google.co.kr/citations?user=cr5wjNYAAAAJ&hl=ko
[2] Jong Chan Kim
https://scholar.google.co.kr/citations?user=YGPImkUAAAAJ&hl=ko
[3] Jae-Sub Ko
https://scholar.google.co.kr/citations?user=Vph8kvsAAAAJ&hl=ko
It was our intention to validate the proposed mathematical theories by performing simulations. At the same time, to cover the drawbacks, the contribution parts have been included in every possible section while correcting the contents with the help of a native English speaker to improve readability within a limited time frame. It seems that the Special Issue ‘Sustainable Building Retrofit and Energy Optimization[4]’, to which we’ve submitted our study, was to provide some understanding to the Sustainable engineers studying computerbuilding/energy/optimization engineering from the energy optimization point of view. Thus, we’ve included relevant discussions to make the paper more meaningful. The revised or added parts are being highlighted in red for your possible re-review.
We added the disadvantages of the Neural Network and its references according to the opinion of the Review.
ADD 1)
Neural networks use multiple layers as the number of variables increases, which increases the amount of data. In order to process the increased data, a high-performance processor with parallel processing capability is required, and there are no rules for configuring the input-hidden-output layer of the neural network, so it relies heavily on the user's experience of designing. In addition, if sufficient learning is not achieved, there is a problem that the response performance is deteriorated. In recent years, the field of utilizing neural networks is increasing and the amount of data is rapidly increasing accordingly, but the processing power of the processor cannot keep up with it. When a neural network is used in a control system, it causes an increase in the price of the system and has a disadvantage that does not guarantee sufficient stability[55-56].
ADD 2)
[55] Mijwel, M. M. Artificial Neural Networks Advantages and disadvantages. Available online: https://www.linkedin.com/pulse/artificial-neural-networks-advantages-disadvantages-maad-m-mijwel/ (accessed on 31 March 2020)
[56] Donges, N. 4 REASONS WHY DEEP LEARNING AND NEURAL NETWORKS AREN’T ALWAYS THE RIGHT CHOICE. Available online: https://builtin.com/data-science/disadvantages-neural-networks/ (access on 31 March 2020)
Finally, there are a lot of problems with equations in the whole paper so it must be revised its format (how to min in the *.pdf version). Other minor revision must be done at the first time that some abbreviations are employed (like PI) must be defined. The main idea is of interest and really applied in the sustainability topic.
-----
Reply-----
-----
We checked and revised the format of the equation and modified the first abbreviation as a reviewer's opinion.
CHANGE 1)
The method presented in this paper compares the perturbation & observation (P&O) method, which is used for MPPT control, and the proportional integral (PI) controller used the most in the industrial field and analyzes the performance.
CHANGE 2)
The incremental conductance (IncCond) method uses the slope of the power-voltage (P-V) curve, which represents the relationship between the power ( ) and the voltage ( ) of the solar cell [27-36].
ADD 3)
Figure 16 shows the environment for the experiment. Fig. 16 (a) is a circuit for the experiment, and Fig. 16 (b) shows artificial light for constant radiation and PV module. The specifications of the PV module and smartphone battery used in the experiment are shown in Table 2, and radiation by artificial lighting is about 100 .
ADD 4)
(a)
(b)
Figure 16. Environment for experiment
(a) Circuit for experiment; (b) Artificial light and PV module
ADD 5)
Table 2.Specifications of the PV module and Battery
|
|
Parameter |
Value |
|
PV Module |
Maximum Power |
30 [Wp] |
|
Open Circuit Voltage |
21 [V] |
|
|
Optimum operation Voltage |
17.5 [V] |
|
|
Short Circuit Current |
2 [A] |
|
|
Optimum Operation Current |
1.7 [A] |
|
|
Battery |
Type |
Li-ion |
|
Capacity |
3200 [mAh] |
ADD 6)
Abbreviations:
MPPT: Maximum Power Point Tracking
PV: PhotoVoltaic
P&O: Perturbation & Observation
PI: Proportional Integral
MPP: maximum power point
CV: Constant Voltage
STC: Standard Test Condition
OCV: Open Circuit Voltage
IncCond: Incremental Conductance
P-V: Power-Voltage
E: Error
CE: Changing Error
MPP: Maximum Power Point
NB: Negative Big
NM: Negative Medium
NS: Negative Small
PB: Positive Big
PM: Positive Medium
PS: Positive Small
CE: Changing Error

Reviewer 4 Report
Revision of "Optimization Design and Test Bed of Fuzzy Control Rule Base for PV System MPPT in Micro Grid", submitted for publication in Sustainability MDPI
The topic concerns the optimization of the energy conversion from PV systems. Indeed, this depends from the environmental conditions (radiation and temperature) and the aim of the paper is the maximization of the output of energy conversion. In particular, by taking into account the voltage-current relationship, the authors would provide the working of the system at the right operating point. More in detail, here, fuzzy control is used, being suitable because it takes into account nonlinear phenomena. The study is original, well written and clear. My opinion is favorable even if written English must be improved (shorter sentences) and the following suggestions must be considered.
Clarify MPPT in the abstract.
Avoid the use of not-explicated acronyms in the abstract. At the firs use, the entire words have to be reported.
The abstract must be clearer and shorter.
Please, the abstract must provide the main outcomes.
There are some red lines in the paper. Why?
Please, improve the nomenclature (the list of abbreviation). Indeed, many acronyms and symbols are used, but also several electrical parameters. Add these in the nomenclature, with proper units, when necessary.
Please, there are many multiples citations. Please split. Several papers cannot be cited in the same sentences, at the same time. Two or three (maximum) papers can be cited together.
For a clearer understanding, add units on the X and Y axes of figures 2 and 5, 6 and 8.
Line 269: how you have determined the net radiation on PV modules equal to 100 W/m2?
Which kind of artificial lamps have been used and how the efficiency and lighting wavelengths can affect the electricity conversion?
Figures 17-21, add units on the axis.
Please, improve the discussion. Add more comments.
The conclusion section is meaningful but, please, add here also the main quantitative results.
As said, according to me the paper is pleasant and interesting. Well-done. However, minor revisions are requested.
Author Response
Comments and Suggestions for Authors
Revision of "Optimization Design and Test Bed of Fuzzy Control Rule Base for PV System MPPT in Micro Grid", submitted for publication in Sustainability MDPI
The topic concerns the optimization of the energy conversion from PV systems. Indeed, this depends from the environmental conditions (radiation and temperature) and the aim of the paper is the maximization of the output of energy conversion. In particular, by taking into account the voltage-current relationship, the authors would provide the working of the system at the right operating point. More in detail, here, fuzzy control is used, being suitable because it takes into account nonlinear phenomena. The study is original, well written and clear. My opinion is favorable even if written English must be improved (shorter sentences) and the following suggestions must be considered.
Clarify MPPT in the abstract.
Avoid the use of not-explicated acronyms in the abstract. At the firs use, the entire words have to be reported. The abstract must be clearer and shorter. Please, the abstract must provide the main outcomes.
-----
Reply -----
-----
First of all, thank you for reviewing our paper in detail and giving us an appropriate comment.
We revised the abstract according to the reviewer's opinion. The revised or added parts are being highlighted in red for your possible re-review.
CHANGE 1)
Abstract: This paper presents an optimal design of a fuzzy control rule base for tracking the maximum power point of a photovoltaic (PV) system. Fuzzy control is used for maximum power point tracking (MPPT) of PV systems because it has the advantage of processing nonlinear systems. The rule base of fuzzy control depends on the user or designer's experience and determines the fuzzy control’s performance. In this paper, we divide the MPPT state of the PV system into four cases according to the operating conditions, and propose the rule base design of the fuzzy control according to each case. The proposed method in the paper tests the MPPT performance using artificial lighting and compares the results with the conventional control method (PI and P&O method) to prove its effectiveness.
There are some red lines in the paper. Why?
-----
Reply -----
-----
Among the terms used in the paper, the words indicated by the red line are “IncCond, Zadeh, fuzzification defuzzification”. First, IncCond stands for Incremental Conductance. Second, Zadeh is the name of the first researcher to publish fuzzy control theory. Lastly, fuzzification and defuzzification are technical terms used in fuzzy control, and are not defined in the dictionary and are marked with red lines.
Please, improve the nomenclature (the list of abbreviation). Indeed, many acronyms and symbols are used, but also several electrical parameters. Add these in the nomenclature, with proper units, when necessary.
-----
Reply -----
-----
We added abbreviations according to the reviewer's comments.
ADD 1)
Abbreviations:
MPPT: Maximum Power Point Tracking
PV: PhotoVoltaic
P&O: Perturbation & Observation
PI: Proportional Integral
MPP: maximum power point
CV: Constant Voltage
STC: Standard Test Condition
OCV: Open Circuit Voltage
IncCond: Incremental Conductance
P-V: Power-Voltage
: Voltage of PV module
: Current of PV module
: power of PV module
: Changing of
: Changing of
PWM : Pulse Width Modulation
E: Error
CE: Changing Error
NB: Negative Big
NM: Negative Medium
ZE : Zero
NS: Negative Small
PB: Positive Big
PM: Positive Medium
PS: Positive Small
GE : Gain of Error in fuzzy control
GCE : Gain of Changing Error in fuzzy control
GU : Gain of oUtput
: Changing of duty ratio
COG : Center Of Gravity
Please, there are many multiples citations. Please split. Several papers cannot be cited in the same sentences, at the same time. Two or three (maximum) papers can be cited together.
-----
Reply -----
-----
We have revised the references according to the reviewer's opinion.
CHANGE 1)
References
- Kim, J. C.; Huh, J. H.; Ko, J. S. Improvement of MPPT Control Performance Using Fuzzy Control and VGPI in the PV System for Micro Grid. sustainability, 2019, 11, 5891.
- Liu, B.; Li, K.; Niu, D. D.; Liu, Y. The characteristic analysis of the solar energy photovoltaic power generation system. IOP Conference Series: Materials Science and Engineering. 2017, 164, 1-6.
- Fratini, P.; Moretti, E.; Belloni, E. Energy and economic evaluation of solar photovoltaics plants: influence of different input parameters. PROCEEDINGS OF ECOS 2012, PERUGIA, ITALY, 26-29 June 2012; pp. 1-14
- CONSERVE ENERGY FUTURE, What are Alternative Energy Sources?. Available on line: https://www.conserve-energy-future.com/alternativeenergysources.php/ (accessed on 10 February 2020)
- Wang, Y.; Huang, Y.; Wang, Y.; Li, F.; Zhang, Y.; Tian, C. Operation Optimization in a Smart Micro-Grid in the Presence of Distributed Generation and Demand Response. MDPI-sustainability. 2018, 10, 847.
- Huang, H.; Nie, S.; Lin, J.; Wang, Y.; Dong, J. Optimization of Peer-to-Peer Power Trading in a Microgrid with Distributed PV and Battery Energy Storage Systems. MDPI-sustainability. 2020, 12, 923.
- He, M. F.; Zhang, F, X.; Huang, Y.; Chen, J.; Wang, J.; Wang, R. A Distributed Demand Side Energy Management Algorithm for Smart Grid. MDPI-energies. 2019, 12, 426.
- Electrical 4U. Characteristics of a Solar Cell and Parameters of a Solar Cell. Available online: https://www.electrical4u.com/characteristics-and-parameters-of-a-solar-cell/ (accessed on 10 February 2020)
- Masoum, M. A. S.; Dehbonei, H.; Fuchs, E. F. Theoretical and experimental analyses of photovoltaic systems with voltage and current-based maximum power-point tracking. IEEE Trans Energy Convers. 202, 17, 514-522
- Elgendy, M. A.; Zahawi, B.; Atkinson, D. J. Comparison of directly connected and constant voltage controlled photovoltaic pumping systems. IEEE Trans Sustain Energy. 2010, 1, 184-192
- Lasheen, M.; Rahman, A. K.; Abdel-Salam, M.; Ookawara, S. Performance Enhancement of Constant Voltage Based MPPT for Photovoltaic Application Using Genetic Algorithm. 3rd International Conference on Power and Energy Systems Engineering, Kitakyushu, Japan, 8-12 September 2016, 217-222.
- Baimel, D.; Shkoury, R.; Elbaz, L.; Tapuchi, S.; Baimel, N. Novel optimized method for maximum power point tracking in PV systems using Fractional Open Circuit Voltage technique. In Proceedings of the 2016 International Symposium on Power Electronics, Electrical Drives, Automation and Motion, Anacapri, Italy, 22–24 June 2016; pp. 889–894.
- Huang, Y.P. A rapid maximum power measurement system for high-concentration photovoltaic modules using the fractional open-circuit voltage technique and controllable electronic load. IEEE J. Photovolt. 2014, 4, 1610–1617.
- Baimel, D.; Tapuchi, S.; Levron, Y.; Belikov, J. Improved Fractional Open Circuit Voltage MPPT Methods for PV Systems. MDPI-electronics, 2019, 8-321.
- Elgendy, M.A.; Zahawi, B.; Atkinson, D.J. Assessment of perturb and observe MPPT algorithm implementation techniques for PV pumping applications. IEEE Trans. Sustain. Energy 2012, 3, 21–33.
- Raj, C.M.J.S.; Jeyakumar, A.E. A novel maximum power point tracking technique for photovoltaic module based on power plane analysis of I–V characteristics. IEEE Trans. Ind. Electron. 2014, 61, 4734–4745.
- Kollimalla, S.K.; Mishra, M.K. A novel adaptive P&O MPPT algorithm considering sudden changes in the irradiance. IEEE Trans. Energy Convers. 2014, 29, 602–610.
- Hossain, M.J.; Tiwari, B.; Bhattacharya, I. An adaptive step size incremental conductance method for faster maximum power point tracking. In Proceedings of the 2016 IEEE 43rd Photovoltaic Specialists Conference, Portland, OR, USA, 5–10 June 2016; pp. 3230–3233.
- Lakshmi, D.; Rashmi, M. A modified incremental conductance algorithm for partially shaded PV array. In Proceedings of the 2017 International Conference on Technological Advancements in Power and Energy, Kollam, India, 21–23 Decembner 2017; pp. 1–6. Electronics 2019, 8, 321
- Sivakumar, P.; Kader, A.A.; Kaliavaradhan, Y.; Arutchelvi, M. Analysis and enhancement of PV efficiency with incremental conductance MPPT technique under non-linear loading conditions. Renew. Energy 2015, 81, 543–550.
- Koad, R. B. A.; Zobaa, A. F. Comparative study of five maximum power point tracing techniques for photovoltaic systems. International Journal on Energy Conversion, 2014, 2, 17-25
- Bendib, B.; Belmili, Hocine, Krim, F. A survey of the most used MPPT method: Conventional and advanced algorithms applied for photovoltaic systems. Renewable and Sustainable Energy Reviews, 2015, 637-648
- Swain, N.; Panigrahi, C. K.; Ali, S. M. Application of PI and MPPT Controller to DC-DC Converter for Constant Voltage & Power Application. IOSR-JEEE, 2016, 11, 8-15
- Anil, G.; Murugan, N.; Ubaid, M. PI Controller based MPPT for a PV System. IOSR-JEEE, 2013, 6, 10-15.
- Yilmaz, U.; Kircay, A.; Borekci, S. PV system fuzzy logic MPPT method and PI control as a charge controller.
- Algarin, C. R.; Giraldo, J. T.; Alvarez, O. R. Fuzzy Logic Based MPPT Controller for a PV System. MDPI-energies, 2017, 10, 2016.
- Yaqin, E. N.; Abdullah, A. G.; Hakim, D. L.; Nandiyanto, A. B. D. MPPT based on Fuzzy Logic Controller for Photovoltaic System using PSIM and Simulink. IOP Conference Series: Meterials Science and Engineering, 2018, 288.
- Soufi, Y.; Bechouat, M.; Bouallegue, K. Maximum power point tracking fuzzy logic control for photovoltaic system. 2014 International Conference on Renewable Energy Research and Application(ICRERA), 2014 , 902-906.
- Yaichi, M.; Fellah, M. K.; Mammeri, A. A Neural Network Based MPPT Technique Controller for Photovoltaic Pumping System. International Journal of Power Electronics and Drive System, 2014, 4, 241-255.
- Elobaid, L. M.; Abdelsalam, A. K.; Zakzouk, E. E. Artificial neural network-based photovoltaic maximum power point tracking techniques: a servey. IET Renewable Power Generation, 2015, 9, 1043-1063
- Mijwel, M. M. Artificial Neural Networks Advantages and disadvantages. Available online: https://www.linkedin.com/pulse/artificial-neural-networks-advantages-disadvantages-maad-m-mijwel/ (accessed on 31 March 2020)
- Donges, N. 4 REASONS WHY DEEP LEARNING AND NEURAL NETWORKS AREN’T ALWAYS THE RIGHT CHOICE. Available online: https://builtin.com/data-science/disadvantages-neural-networks/ (access on 31 March 2020)
- Dutu, L. C.; Mauris, G.; Bolon, P. A Fast and Accurate Rule-Base Generation method for Mamdani Fuzzy Systems. IEEE Transactions on Fuzzy Systems, 2018, 715-733
- Zadeh, L. A. (1965). Fuzzy sets. Information and Control 8, pp. 338–353.
- B´ardossy, A., and Duckstein, L. (1995). “Fuzzy rule-based modeling with application to geophysical, biological and engineering systems.” CRC Press.
- Chi, Z., Yan, H., and Pham, T. (1996). “Fuzzy algorithms: with applications to image processing and pattern recognition.” World Scientific.
- Hirota, K. (Ed.) (1993). “Industrial applications of fuzzy technology.” Springer-Verlag.
- Pedrycz, W. (Ed.) (1996). “Fuzzy Modelling. Paradigms and Practice.” Kluwer Academic Press.
- Coelho, R.F.; Martins, D.C. An Optimized Maximum Power Point Tracking Method Based on PV Surface Temperature Measurement. Available online: https://www.intechopen.com/books/sustainable-energy-recent-studies/an-optimized-maximum-power-point-tracking-method-based-on-pv-surface-temperature-measurement/ (accessed on 10 February 2020).
For a clearer understanding, add units on the X and Y axes of figures 2 and 5, 6 and 8.
-----
Reply -----
-----
We added units to the X and Y axes as reviewer comments.
CHANGE 1)
Figure 2. Maximum Power Point Tracking direction according to error (E).
Figure 3. Maximum power point tracking characteristics (left side of maximum power point).
Figure 5. Maximum power point tracking characteristics (right side of maximum power point).
Line 269: how you have determined the net radiation on PV modules equal to 100 W/m2?
-----
Reply -----
-----
We added related content according to the reviewer's opinion.
ADD 1)
The radiation from the artificial light source was calculated using the output ratio. The rated output of the PV module used for the experiment is 30 [W], which is a value measured at radiation 1000 . In the paper, the output measured through the experiment using artificial lighting is about 3 [W], and when calculated proportionally, the radiation is about 100 .
Which kind of artificial lamps have been used and how the efficiency and lighting wavelengths can affect the electricity conversion?
-----
Reply -----
-----
We added related content according to the reviewer's opinion.
ADD 1)
In this paper, artificial lighting is used to test the MPPT performance of PV systems. Artificial lighting has lower radiation than sunlight, and for that reason, it produces very low power compared to the rated power [40]. And the output of the PV system is proportional to the amount of solar radiation when the temperature is constant [40-41]. Artificial lighting can maintain a constant radiation and can be appropriately changed by the user. Therefore, many studies using artificial lighting have been proposed to test the MPPT performance of PV systems in various environments [42-43]. The artificial light used in the paper is an incandescent lamp, and the incandescent lamp is the most suitable artificial light source for solar applications, especially in the case of crystalline silicon solar cells, the performance is improved over the AM1.5 spectrum [44].
ADD 2)
Reference
- W. Xiao and W. G. Dunford, "Evaluating maximum power point tracking performance by using artificial lights," 30th Annual Conference of IEEE Industrial Electronics Society, 2004. IECON 2004, Busan, South Korea, 2004, pp. 2883-2887 Vol. 3.
- Zandi, Z., Mazinan, A.H. Maximum power point tracking of the solar power plants in shadow mode through artificial neural network. Complex Intell. Syst. 2019, 5,315–330
- Nimje, P.; Nirali, S.; Saxena, A. Energy Retrofitting: Design of Lighting and Solar Photovoltaic System. International Journal of Applied Engineering Research. vol. 10, no. 57, 2015, pp. 148-153
- A. S. Weddell, G. V. Merrett and B. M. Al-Hashimi, "Ultra low-power photovoltaic MPPT technique for indoor and outdoor wireless sensor nodes," 2011 Design, Automation & Test in Europe, Grenoble, 2011, pp. 1-4.
- Minnaert, B.; Veelaert, P. A Proposal for Typical Artificial Light Sources for the Characterization of Indoor Photovoltaic Applications. Energies 2014, 7, 1500-1516.
Figures 17-21, add units on the axis.
-----
Reply -----
-----
We revised the figures with missing axis units according to the reviewer's opinion.
CHANGE 1)
(d)
Figure 14. MPPT control characteristics (proposed Fuzzy MPPT method)
(d) Duty Ratio and
(b)
Figure 15. Control characteristics according to the error and changing error of the proposed Fuzzy MPPT method
(b) Region A Expansion
(d)
Figure 16. MPPT control characteristics (PI MPPT)
(d) Duty Ratio and
(d)
Figure 17. MPPT control characteristics (P&O MPPT)
(d) Duty Ratio and
Please, improve the discussion. Add more comments.
Reply à
We revised the abstract according to the review comments, and added a description of the experimental environment. And the conclusion also added quantitative results.
CHANGE 1)
Abstract: This paper presents an optimal design of a fuzzy control rule base for tracking the maximum power point of a photovoltaic (PV) system. Fuzzy control is used for maximum power point tracking (MPPT) of PV systems because it has the advantage of processing nonlinear systems. The rule base of fuzzy control depends on the user or designer's experience and determines the fuzzy control’s performance. In this paper, we divide the MPPT state of the PV system into four cases according to the operating conditions, and propose the rule base design of the fuzzy control according to each case. The proposed method in the paper tests the MPPT performance using artificial lighting and compares the results with the conventional control method (PI and P & O method) to prove its effectiveness.
CHANGE 2)
When the radiation is constant, the error values in the steady state are 81.51 [mW], 199.09 [mW] and 281.74 [mW] for the proposed method, conventional PI method, and P & O method, respectively. The proposed method has the lowest error and the highest average power. The steady-state error was shown in the order of the proposed method <PI method <P&O method, even with increasing radiation conditions, and the tracking accuracy of the proposed method was the highest.
ADD 1)
In this paper, artificial lighting is used to test the MPPT performance of PV systems. Artificial lighting has lower radiation than sunlight, and for that reason, it produces very low power compared to the rated power [40]. And the output of the PV system is proportional to the amount of solar radiation when the temperature is constant [40-41]. Artificial lighting can maintain a constant radiation and can be appropriately changed by the user. Therefore, many studies using artificial lighting have been proposed to test the MPPT performance of PV systems in various environments [42-43]. The artificial light used in the paper is an incandescent lamp, and the incandescent lamp is the most suitable artificial light source for solar applications, especially in the case of crystalline silicon solar cells, the performance is improved over the AM1.5 spectrum [44].
ADD 2)
Reference
- W. Xiao and W. G. Dunford, "Evaluating maximum power point tracking performance by using artificial lights," 30th Annual Conference of IEEE Industrial Electronics Society, 2004. IECON 2004, Busan, South Korea, 2004, pp. 2883-2887 Vol. 3.
- Zandi, Z., Mazinan, A.H. Maximum power point tracking of the solar power plants in shadow mode through artificial neural network. Complex Intell. Syst. 2019, 5,315–330
- Nimje, P.; Nirali, S.; Saxena, A. Energy Retrofitting: Design of Lighting and Solar Photovoltaic System. International Journal of Applied Engineering Research. vol. 10, no. 57, 2015, pp. 148-153
- A. S. Weddell, G. V. Merrett and B. M. Al-Hashimi, "Ultra low-power photovoltaic MPPT technique for indoor and outdoor wireless sensor nodes," 2011 Design, Automation & Test in Europe, Grenoble, 2011, pp. 1-4.
The conclusion section is meaningful but, please, add here also the main quantitative results. As said, according to me the paper is pleasant and interesting. Well-done. However, minor revisions are requested.
Reply à
We revised it by adding quantitative results to the conclusion.
CHANGE 1)
When the radiation is constant, the error values in the steady state are 81.51 [mW], 199.09 [mW] and 281.74 [mW] for the proposed method, conventional PI method, and P & O method, respectively. The proposed method has the lowest error and the highest average power. The steady-state error was shown in the order of the proposed method <PI method <P&O method, even with increasing radiation conditions, and the tracking accuracy of the proposed method was the highest.
Round 2
Reviewer 2 Report
The paper is too long. Most readers should have enough knowledge of MPPT for PV systems, and some part of the paper can be simplified.
The experimental system need to be improved. the system power rating is not enough to validate the effectiveness of the method. a higher power rating experimental system is expected.
The "Environment for experiment" design has problems, and it can not generate a high and uniform irradiance for PV. the PV output current is low, and may have multiple maximum power points.
authors need to explain why to choose smart phone battery? what is the battery voltage?
In the experimental results, the PWM waveform is confusing, authors need to explain.
Author Response
Comments and Suggestions for Authors
The paper is too long. Most readers should have enough knowledge of MPPT for PV systems, and some part of the paper can be simplified.
-----
Reply -----
-----
First of all, thank you for reviewing our paper in detail and giving us an appropriate comment.
We reduced the length of the paper by deleting the "2. Conventional MPPT method" according to the reviewer's opinion.
The revised or added parts are being highlighted in red for your possible re-review.
CHANGE 1)
- Conventional MPPT method
Representative methods for the MPPT control of PV systems include the P&O and IncCond methods [1]. Figure 1 shows the flowchart of the P&O method, which observes the change in power by changing the voltage and controls the direction in which the power continues to increase. When the voltage increases, it increases as the power increases; when the voltage decreases, it continues to decrease as the power increases. If the power is reduced in response to the change in voltage, the control method allows the power to increase again by reversing the direction of change of the voltage.
Figure 1. Flowchart of the P&O method.
Figure 2 shows the relationship between the voltage ( ) and power ( ) of the PV system. The slope of power varies depending on the operating point. Figure 3 shows the IncCond method using the slope characteristics of the P-V curve. If the power slope is positive, increase the voltage. If the slope is negative, decrease the voltage to track the maximum power point.
Figure 2. P-V curve of the PV system.
Figure 3. Flowchart of the IncCond method.
The experimental system need to be improved. The system power rating is not enough to validate the effectiveness of the method. A higher power rating experimental system is expected.
-----
Reply -----
-----
We have added content and reference about the effectiveness of MPPT performance verification of PV systems using artificial lighting.
ADD 1)
In this paper, artificial lighting is used to test the MPPT performance of PV systems. Artificial lighting has lower radiation than sunlight, and for that reason, it produces very low power compared to the rated power [40]. And the output of the PV system is proportional to the amount of solar radiation when the temperature is constant [40-41]. Artificial lighting can maintain a constant radiation and can be appropriately changed by the user. Therefore, many studies using artificial lighting have been proposed to test the MPPT performance of PV systems in various environments [42-43]. The artificial light used in the paper is an incandescent lamp, and the incandescent lamp is the most suitable artificial light source for solar applications, especially in the case of crystalline silicon solar cells, the performance is improved over the AM1.5 spectrum [44].
ADD 2)
Reference
- W. Xiao and W. G. Dunford, "Evaluating maximum power point tracking performance by using artificial lights," 30th Annual Conference of IEEE Industrial Electronics Society, 2004. IECON 2004, Busan, South Korea, 2004, pp. 2883-2887 Vol. 3.
- Zandi, Z., Mazinan, A.H. Maximum power point tracking of the solar power plants in shadow mode through artificial neural network. Complex Intell. Syst. 2019, 5,315–330
- Nimje, P.; Nirali, S.; Saxena, A. Energy Retrofitting: Design of Lighting and Solar Photovoltaic System. International Journal of Applied Engineering Research. vol. 10, no. 57, 2015, pp. 148-153
- A. S. Weddell, G. V. Merrett and B. M. Al-Hashimi, "Ultra low-power photovoltaic MPPT technique for indoor and outdoor wireless sensor nodes," 2011 Design, Automation & Test in Europe, Grenoble, 2011, pp. 1-4.
- Minnaert, B.; Veelaert, P. A Proposal for Typical Artificial Light Sources for the Characterization of Indoor Photovoltaic Applications. Energies 2014, 7, 1500-1516.
The "Environment for experiment" design has problems, and it can not generate a high and uniform irradiance for PV. PV output current is low, and may have multiple maximum power points.
-----
Reply -----
-----
Figure 15(c) shows the P-V curve by the Fuzzy MPPT method presented in the paper, and Figure 19 shows the steady-state response characteristics in the P-V curve of the fuzzy MPPT method, PI MPPT method and P & O MPPT method. In Figures 15 and 19, all controllers are controlled to track one maximum power point.
(c)
Figure 15. Control characteristics according to the error and changing error of the proposed Fuzzy MPPT method
(a)
(b)
(c)
Figure 19. Comparison of MPPT response characteristics in steady state
(a) Proposed fuzzy MPPT method; (b) PI MPPT method; (c) P&O MPPT method
Authors need to explain why to choose smart phone battery? What is the battery voltage?
-----
Reply -----
-----
We added the reason for using the battery of the mobile phone and the voltage of the battery.
ADD 1)
Mobile phones have a very close relationship with everyday life, and recently PV systems are also used to provide charging of mobile phones in places such as public places and bus stops. Therefore, in the paper, the battery of the mobile phone was used as the load of the PV system.
ADD 2)
Table 2.Specifications of the PV module and Battery
|
|
Parameter |
Value |
|
PV Module |
Maximum Power |
30 [Wp] |
|
Open Circuit Voltage |
21 [V] |
|
|
Optimum operation Voltage |
17.5 [V] |
|
|
Short Circuit Current |
2 [A] |
|
|
Optimum Operation Current |
1.7 [A] |
|
|
Battery
|
Type |
Li-ion |
|
Capacity |
3200 [mAh] |
|
|
Voltage |
3.7[V] |
In the experimental results, the PWM waveform is confusing, authors need to explain.
-----
Reply -----
-----
We modified the PWM waveform to Duty Ratio.
CHANGE 1)
(d)
Figure 14. MPPT control characteristics (proposed Fuzzy MPPT method)
(d) Duty Ratio and
(d)
Figure 16. MPPT control characteristics (PI MPPT)
(d) Duty Ratio and
(d)
Figure 17. MPPT control characteristics (P&O MPPT)
(d) Duty Ratio and